# Open-source, Python-based, hardware and software for controlling behavioural neuroscience experiments

**Thomas Akam[1,2]\*, Andy Lustig[3], James M Rowland[4], Sampath KT Kapanaiah[5], Joan Esteve-Agraz[6], Mariangela Panniello[4,7], Cristina Márquez[6], Michael M Kohl[4,7], Dennis Kätzel[5], Rui M Costa[2,8†], Mark E Walton[1,9†]**

[1]Department of Experimental Psychology, University of Oxford, Oxford, United Kingdom; [2]Champalimaud Neuroscience Program, Champalimaud Centre for the Unknown, Lisbon, Portugal; [3]Janelia Research Campus, Howard Hughes Medical Institute, Ashburn, United States; [4]Department of Physiology Anatomy & Genetics, University of Oxford, Oxford, United Kingdom; [5]Institute of Applied Physiology, Ulm University, Ulm, Germany; [6]Instituto de Neurociencias (Universidad Miguel Hernández-Consejo Superior de Investigaciones Científicas), Sant Joan d'Alacant, Spain; [7]Institute of Neuroscience and Psychology, University of Glasgow, Glasgow, United Kingdom; [8]Department of Neuroscience and Neurology, Zuckerman Mind Brain Behavior Institute, Columbia University, New York, United States; [9]Wellcome Centre for Integrative Neuroimaging, University of Oxford, Oxford, United Kingdom

**\*For correspondence:**
thomas.akam@psy.ox.ac.uk

†These authors contributed equally to this work

**Abstract** Laboratory behavioural tasks are an essential research tool. As questions asked of behaviour and brain activity become more sophisticated, the ability to specify and run richly structured tasks becomes more important. An increasing focus on reproducibility also necessitates accurate communication of task logic to other researchers. To these ends, we developed pyControl, a system of open-source hardware and software for controlling behavioural experiments comprising a simple yet flexible Python-based syntax for specifying tasks as extended state machines, hardware modules for building behavioural setups, and a graphical user interface designed for efficiently running high-throughput experiments on many setups in parallel, all with extensive online documentation. These tools make it quicker, easier, and cheaper to implement rich behavioural tasks at scale. As important, pyControl facilitates communication and reproducibility of behavioural experiments through a highly readable task definition syntax and self-documenting features. Here, we outline the system's design and rationale, present validation experiments characterising system performance, and demonstrate example applications in freely moving and head-fixed mouse behaviour.

## Editor's evaluation

The importance of carefully-considered animal behavior to systems neuroscience cannot be overstated. Despite this, flexible tools for carefully monitoring and controlling behavioral apparatuses have often required significant new development by individual laboratories. The open source pyControl software and hardware toolbox is an excellent exemplar of a robust and reliable platform for experiments, with a simple interface, good performance, excellent documentation, and a growing an engaged user community. This work benchmarks and documents pyControl and hopefully will serve as a useful introduction to an even broader community.

## Introduction

Animal behaviour is of fundamental scientific interest, both in its own right and in relation to brain function (*Krakauer et al., 2017*). Though understanding natural behaviour is the ultimate goal, the tight control offered by laboratory tasks remains an essential tool in characterising learning mechanisms. To serve the needs of contemporary neuroscience, hardware and software for controlling behavioural experiments should be both flexible and easy to use. Additionally, an increasing focus on reproducibility (*Baker, 2016*; *International Brain Laboratory et al., 2021*) necessitates that behaviour control systems facilitate communication and replication of behavioural paradigms across labs.

Available commercial solutions often fall short of these desiderata. Proprietary closed-source hardware and software make it difficult to extend or adapt functionality beyond explicitly implemented use cases. Additionally, programming behavioural tasks on commercial systems can be surprisingly non-user-friendly, perhaps due to limitations of underlying legacy hardware. Commercial hardware is also typically very expensive considering the level of technology it represents, disadvantaging researchers outside well-funded institutions (*Marder, 2013*; *Maia Chagas, 2018*), and constraining the ability to scale behavioural assays for high throughput.

For these reasons, many groups implement their own behavioural hardware either using low-cost microcontrollers such as Arduinos or raspberry PI, or generic laboratory control software such as Labview (*Devarakonda et al., 2016*; *O'Leary et al., 2018*; *Gurley, 2019*; *Bhagat et al., 2020*; *Buscher et al., 2020*). Though highly flexible, building behavioural control systems from scratch has some disadvantages. It results in much duplication of effort as a lot of the required functionality is generic across experiments. Additionally, unless custom systems are well documented, it is hard for users to meaningfully share experimental protocols. This is important because scientific publications do not consistently contain sufficient information to constrain the details of the task used, yet such details are often crucial for reproducing the behaviour. Making task code public is therefore key to reproducibility, but this is only effective if it is readable and documented, as well as functional.

To address these limitations, we developed *pyControl*; a system of open-source hardware and software for controlling behavioural experiments. We report the design and rationale of system components, validation experiments characterising system performance, and behavioural data illustrating applications in three widely used, contrasting behavioural paradigms: the 5-choice serial reaction time task (5-CSRTT) in operant chambers, sensory discrimination in head-fixed animals, and a social decision-making task in a maze apparatus.

## Results

### System overview

pyControl consists of three components, the pyControl framework, hardware, and graphical user interface (GUI). The framework implements the syntax used to program behavioural tasks. User-created task definition files, written in Python, run directly on microcontroller hardware, supported by framework code that determines when user-defined functions are called. This takes advantage of Micro-Python, a recently developed port of the popular high-level language Python to microcontrollers. The framework handles functionality that is common across tasks, such as monitoring inputs, setting and checking timers, and streaming data back to the computer. This minimises boilerplate code in task files, while ensuring that common functionality is implemented reliably and efficiently. Combined with Python's highly readable syntax, this results in task files that are quick and straightforward to write, but also easy to read and understand (*Figure 1*), promoting replicability and communication of behavioural experiments.

pyControl hardware consists of a breakout board which interfaces a pyboard microcontroller with ports and connectors, and a set of devices such as nose-pokes, audio boards, LED drivers, rotary encoders, and stepper motor controllers that are connected to the breakout board to create behavioural setups. Breakout boards connect to the computer via USB. Multiple breakout boards can be connected to a single computer, each controlling a separate behavioural setup. pyControl implements a simple but robust mechanism for synchronising data with other systems such as cameras or physiology hardware. All hardware is fully open source, and assembled hardware is available at low cost from the Open Ephys store and LabMaker.

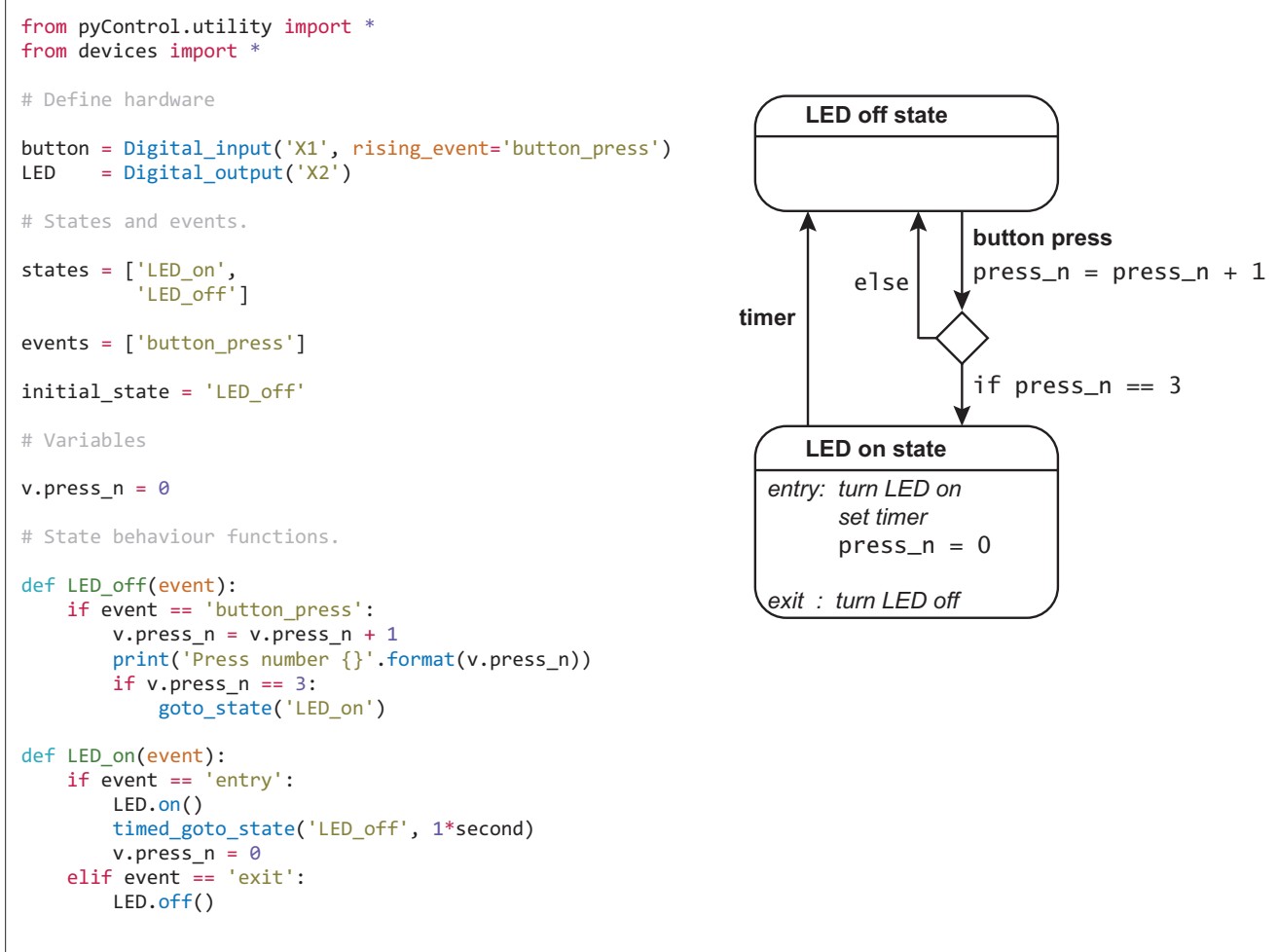

**Figure 1.** Example task. Complete task definition code (left panel) and corresponding state diagram (right panel) for a simple task that turns an LED on for 1 s when a button is pressed three times. Detailed information about the task definition syntax is provided in the Programming Tasks documentation.

The online version of this article includes the following figure supplement(s) for figure 1:

**Figure supplement 1.** Example data file.

The GUI provides a graphical interface for setting up and running experiments, visualising behaviour, and configuring setups, and is designed to facilitate high-throughput behavioural testing on many setups in parallel. To promote replicability, the GUI implements self-documenting features which ensure that all task files used to generate data are stored with the data itself, and that any changes to task parameters from default values are recorded in the data files.

## Task definition syntax

Here, we give an overview of the task definition syntax and how this contributes to the flexibility of the system. Detailed information about task programming is provided in the documentation and set of example tasks is included with the GUI, including probabilistic reversal learning and random ratio instrumental conditioning.

pyControl tasks are implemented as state machines, the basic elements of which are states and events. At any given time, the task is in one of the states, and the current state determines how the task responds to events. Events may be generated externally, for example, by the subject's actions, or internally by timers.

*Figure 1* shows the complete task definition code and the corresponding state diagram for a simple task in which pressing a button three times turns on an LED for 1 s. The code first defines the

hardware that will be used, lists the task's state and event names, specifies the initial state, and initialises task variables.

The code then specifies task behaviour by defining a *state behaviour function* for each state. Whenever an event occurs, the state behaviour function for the current state is called with the event name as an argument. Special events called *entry* and *exit* occur when a state is entered and exited allowing actions to be performed on state transitions. State behaviour functions typically comprise a set of *if* and *else if* statements that determine what happens when different events occur in that state. Any valid MicroPython code can be placed in a state behaviour function, the only constraint being that it must execute fast as it will block further state machine behaviour while executing. Users can define additional functions and classes in the task definition file that can be called from state behaviour functions. For example, code implementing a reversal learning task's block structure might be separated from the state machine code in a separate function, improving readability and maintainability.

As should be clear from the above, while pyControl makes it easy to specify state machines, tasks are not strict finite state machines, in which the response to an event depends *only* on the current state, but rather extended state machines in which variables and arbitrary code can also determine behaviour.

We think this represents a good compromise between enforcing a specific structure on task code, which promotes readability and reliability and allows generic functionality to be efficiently implemented by the framework, while allowing users enough flexibility to compactly define a diverse range of complex tasks.

A key framework component is the ability to set timers to trigger state transitions or events. The *timed_goto_state* function, used in the example, triggers a transition to a specified state after a specified delay. Other functions allow timers to trigger a specified event after a specified delay, or to cancel, pause and un-pause timers that have already been set.

To make things happen in parallel with the main state set of the task, the user can define an *all_states* function which is called, with the event name as an argument, whenever an event occurs irrespective of the state the task is in. This can be used in combination with timers and variables to implement task behaviour that occurs independently from or interacts with the main state set. For example to make something happen after a specified duration, irrespective of the current state, the user can set a timer to trigger an event after the required duration and use the *all_states* function to perform the required action whenever the event occurs.

pyControl provides a set of functions for generating random variables, and maths functions are available via the MicroPython maths module. Though MicroPython implements a large subset of the core Python language (see the MicroPython docs), it is not possible to use packages such as *NumPy* or *SciPy* as they are too large to fit on a microcontroller.

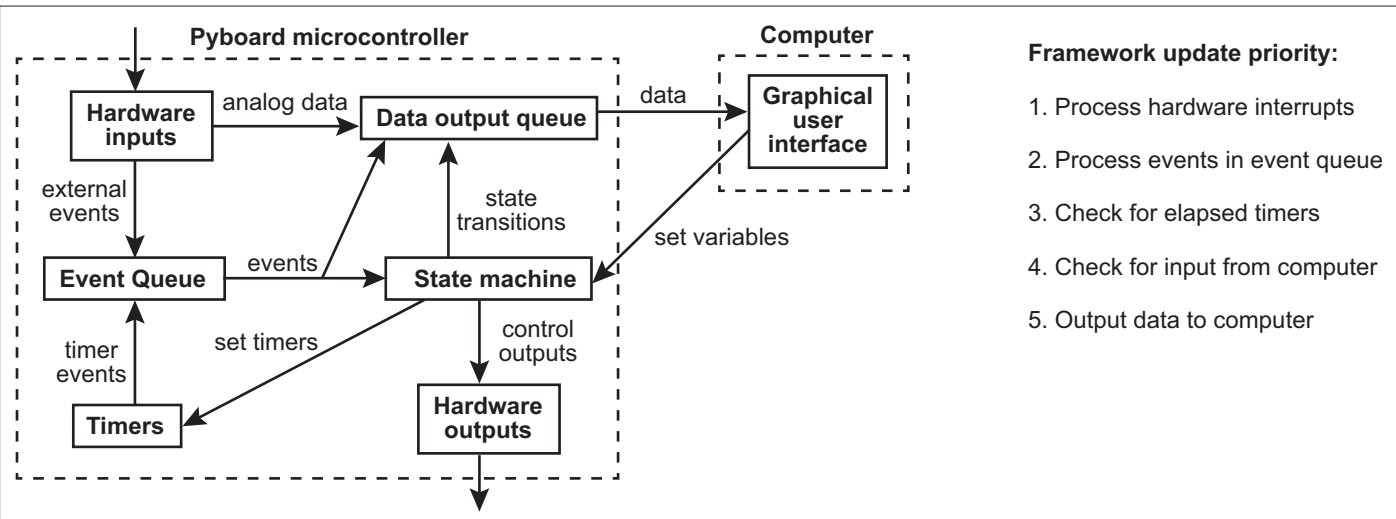

**Figure 2.** Framework diagram. Diagram showing the flow of information between different components of the framework and the graphical user interface (GUI) while a task is running. Right panel shows the priority with which processes occur in the framework update loop.

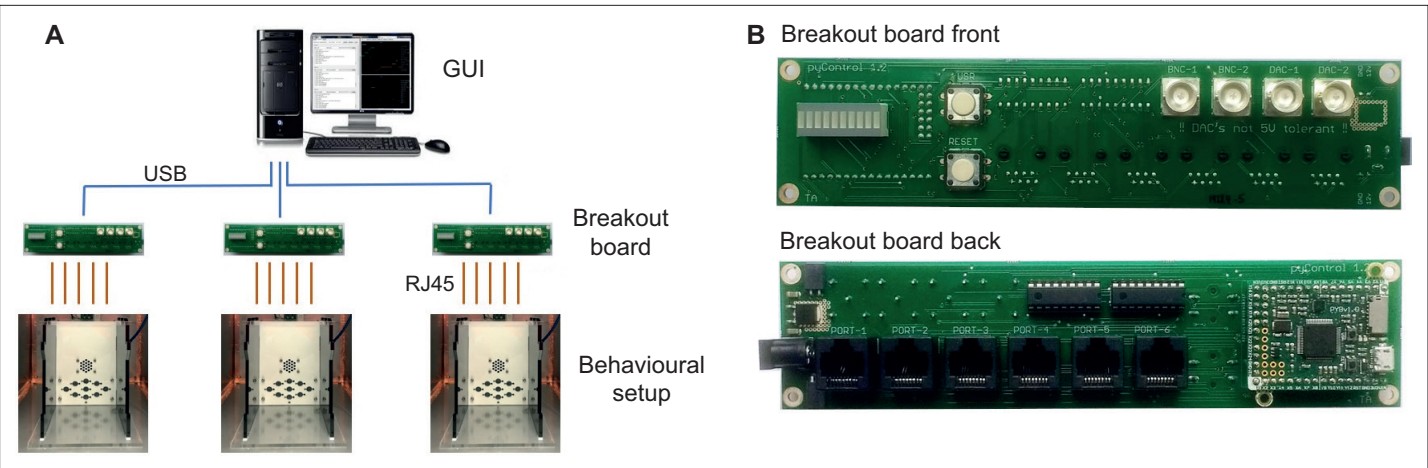

**Figure 3.** pyControl hardware. (**A**) Diagram of a typical pyControl hardware setup, a single computer connects to multiple breakout boards, each of which controls one behavioural setup. Each behavioural setup comprises devices connected to the breakout board RJ45 behaviour ports using standard network cables. (**B**) Breakout board interfacing the pyboard microcontroller with a set of behaviour ports, BNC connectors, indicator LEDs, and user buttons. See *Figure 6—figure supplement 1*, *Figure 7—figure supplement 1*, and *Figure 8—figure supplement 1* for hardware configurations used in the behavioural experiments reported in this article, along with their associated hardware definition files. For more information, see the hardware docs.

## Framework implementation

The pyControl framework consists of approximately 1000 lines of Python code. *Figure 2* shows a simplified diagram of information flow between system components. Hardware inputs and elapsing timers place events in a queue where they await processing by the state machine. When events are processed, they are placed in a data output queue along with any state transitions and user print statements that they generate. This design allows different framework update processes to be prioritised by urgency, rather than by the order in which they become necessary, ensuring the framework responds at low latency even under heavy load (see validation experiments below). Top priority is given to processing hardware interrupts, secondary priority to passing events from the event queue to the state machine and processing their consequences, lowest priority to sending and receiving data from the computer.

Digital inputs are detected by hardware interrupts and can be configured to generate separate framework events on rising and/or falling edges. Analog inputs can stream continuous data to the computer and trigger framework events when the signal goes above and/or below a specified threshold.

## Hardware

A typical pyControl hardware setup consists of a computer running the GUI, connected via USB to one or more breakout boards, each of which controls a single behavioural setup (*Figure 3A*). As task code runs on the microcontroller, the computer does not need to be powerful. We typically use standard office desktops running Windows. We have not systematically tested the maximum number of setups that can be controlled from one computer but have run 24 in parallel without issue.

The breakout board interfaces a pyboard microcontroller (an Arm Cortex M4 running at 168 MHz with 192 KB RAM) with a set of *behaviour ports* used to connect devices that make up behavioural setups, and BNC connectors, indicator LEDs, and user pushbuttons (*Figure 3B*). Each behaviour port is an RJ45 connector (compatible with standard network cables) with power lines (ground, 5 V, 12 V), two digital inputs/output (DIO) lines that are directly connected to microcontroller pins, and two driver lines for switching higher current loads. The driver lines are low-side drivers (i.e. they connect the negative side of the load to ground) that can switch currents up to 150 mA at voltages up to 12 V, with clamp diodes to the 12 V rail to support inductive loads such as solenoids. Two ports have an additional driver line and two have an additional DIO. Six of the behaviour port DIO lines can alternatively be used as analog inputs and two as analog outputs. Three ports support UART and two support I2C serial communication over their DIO lines. The pinout of the behaviour port is detailed in *Table 1*.

**Table 1.** Behaviour port pinout.

All behaviour ports support the standard function for each pin, comprising two digital input/output (DIO) lines connected directly to microcontroller pins, two power driver lines connected to low-side MOSFET drivers for switching higher power loads, and +12 V, + 5 V and ground lines. Some behaviour ports support alternate functions on some pins. On breakout board version 1.2, ports 1 and 2 have an additional power driver line (POW C) and ports 3 and 4 have an additional DIO line (DIO C). Some DIO lines support analog input/output (ADC/DAC), serial communication (I2C, UART, or CAN), or decoding of quadrature signals from rotary encoders (ENC).

**Pinout of behaviour port RJ45 connectors**

| Standard function | Alternate function | Pin |
|---|---|---|
| Ground | None | 2 |
| +5 V | None | 6 |
| +12 V | None | 8 |
| Digital input/output (DIO) A | Analog input (ADC), I2C-SCL, UART-TX, CAN-RX, ENC | 1 |
| Digital input/output (DIO) B | Analog input (ADC), I2C-SDA, UART-RX, CAN-TX, ENC | 4 |
| Power driver (POW) A | None | 3 |
| Power driver (POW) B | None | 7 |
| None | DIO C, POW C, analog output (DAC), analog input (ADC) | 5 |

**Alternate functions available on each behaviour port of breakout board version 1.2**

| Port | Alternate functions |
|---|---|
| 1 | POW C, UART 4, ENC 5, ADC (on DIO A and B) |
| 2 | POW C, CAN 1 |
| 3 | DIO C, DAC 1, I2C 1, UART 1, ENC 4, ADC (on DIO C) |
| 4 | DIO C, DAC 2, I2C 2, UART 3, ADC (on DIO C) |
| 5 | CAN 2 |
| 6 | ADC (on DIO A and B) |

A variety of devices have been developed that connect to the ports, including nose-pokes, levers, audio boards, rotary encoders, stepper motor drivers, lickometers, and LED drivers (*Figure 6—figure supplement 1*, *Figure 7—figure supplement 1*, and *Figure 8—figure supplement 1*). Each has its own driver file that defines a Python class for controlling the device. For detailed information about devices, see the hardware docs. The hardware repository also contains open-source designs for operant boxes and sound attenuating chambers.

Though it is possible to specify the hardware that will be used directly in a task file as shown in *Figure 1*, it is typically done in a separate hardware definition file that is imported by the task. This avoids redundancy when many tasks are run on the same setup. Additionally, abstracting devices used in a task from the specific pins/ports they are connected to allows the same task to run on different setups as long as their hardware definitions instantiate the required devices. See *Figure 6—figure supplement 1*, *Figure 7—figure supplement 1*, and *Figure 8—figure supplement 1* for hardware definitions and corresponding hardware diagrams for the example applications detailed below.

The design choice of running tasks on a microcontroller, and the specific set of devices developed to date, imposes some constraints on experiments supported by the hardware. The limited computational resources preclude generating complex visual stimuli, making pyControl unsuitable for most visual physiology in its current form. The devices for playing audio are aimed at general behavioural neuroscience applications and may not be suitable for some auditory neuroscience applications. One uses the pyboard's internal DAC for stimulus generation, and hence is limited to simple sounds such as sine waves or noise. Another plays WAV files from an SD card, allowing for diverse stimuli but limited to 44 kHz sample rate.

To extend the functionality of pyControl to application not supported by the existing hardware, it is straightforward to interface setups with user-created or commercial devices. This requires creating an

electrical connection between the devices and defining the inputs and outputs in the hardware definition. Triggering external hardware from pyControl, or task events from external devices, is usually achieved by connecting the device to a BNC connector on the breakout board, and using the standard pyControl digital input or output classes. More complex interactions with external devices may involve multiple inputs and outputs and/or serial communication. In this case, the electrical connection is typically made to a behaviour port as these carry multiple signal lines. A port adapter board, which breaks out an RJ45 connector to a screw terminal, simplifies connecting wires. Alternatively, if more complex custom circuitry is required, for example, to interface with a sensor, it may make sense to design a custom-printed circuit board with an RJ45 connector, similar to existing pyControl devices, as this is more scalable and robust than implementing the circuit on a breadboard. To simplify instantiating devices comprising multiple inputs and outputs, or controlling devices which require dedicated code, users can define a Python class representing the device. These are typically simple classes which instantiate the relevant pyControl input and output objects as attributes, and may have methods containing code for controlling the device, for example, to generate serial commands. More information is provided in the hardware docs, and the design files and associated code for existing pyControl devices provide a useful starting point for new designs. Alla Karpova's lab at Janelia Research Campus has independently developed and open-sourced several pyControl-compatible devices (GitHub; *Karpova, 2021*).

For neuroscience applications, straightforward and failsafe synchronisation between behavioural data and other hardware such as cameras or physiology recordings is essential. pyControl implements a simple but robust method for this. Sync pulses are sent from pyControl to the other systems, which each record the pulse times in their own reference frame. The pulse train has random inter-pulse intervals which ensures a unique match between pulse sequences recorded on each system, so it is always possible to identify which pulse corresponds to which even if pulses are missing (e.g. due to forgetting to turn a system on until after the start of a session). This also makes it unambiguous whether two files come from the same session in the event of a file name mix-up. A Python module is provided for converting times between different systems using the sync pulse times recorded by each. For more information, see the synchronisation docs.

## Graphical user interface

The GUI provides two ways of setting up and running tasks; the *Run task* and *Experiments* tabs, as well as a *Setups* tab used to name and configure hardware setups.

The *Run task* tab allows the user to quickly upload and run a task on a single setup. It is typically used for prototyping tasks and testing hardware, but can also be used to acquire data. The values of task variables can be modified before the task is started or while the task is running. During the run, a log of events, state entries, and user print statements is displayed, and the events, states, and any analog signals are plotted live in scrolling plot panels.

The *Experiments* tab is used for running experiments on multiple setups in parallel and is designed to facilitate high-throughput experiments where multiple users run cohorts of animals through a set of boxes. An experiment consists of a set of subjects run in parallel on the same task. If different subjects need to be run in parallel on different tasks, this can be achieved by opening multiple instances of the GUI.

To configure an experiment, the user specifies which subjects will run on which setups, and the values of any variables that will be modified before the task starts. Variables can be set to the same value for all subjects or for individual subjects. Variables can be specified as *Persistent*, causing their value to be stored on the computer at the end of the session, and subsequently set to the same value the next time the experiment is run. Variables can be specified as *Summary,* causing their values to be displayed in a table at the end of the framework run and copied to the clipboard in a format that can be pasted directly into a spreadsheet, for example, to record the number of trials and rewards for each subject. Experiment configurations can be saved and subsequently loaded.

When an experiment is run, the experiments tab changes from the *configure experiment* interface to a *run experiment* interface. The session can be started and stopped individually for each subject or simultaneously for all subjects. While each setup is running, a log of events, state entries, and user print statements is displayed, along with the current state, most recent event, and print statement (*Figure 4*). Variable values can be viewed and modified for individual subjects during the session.

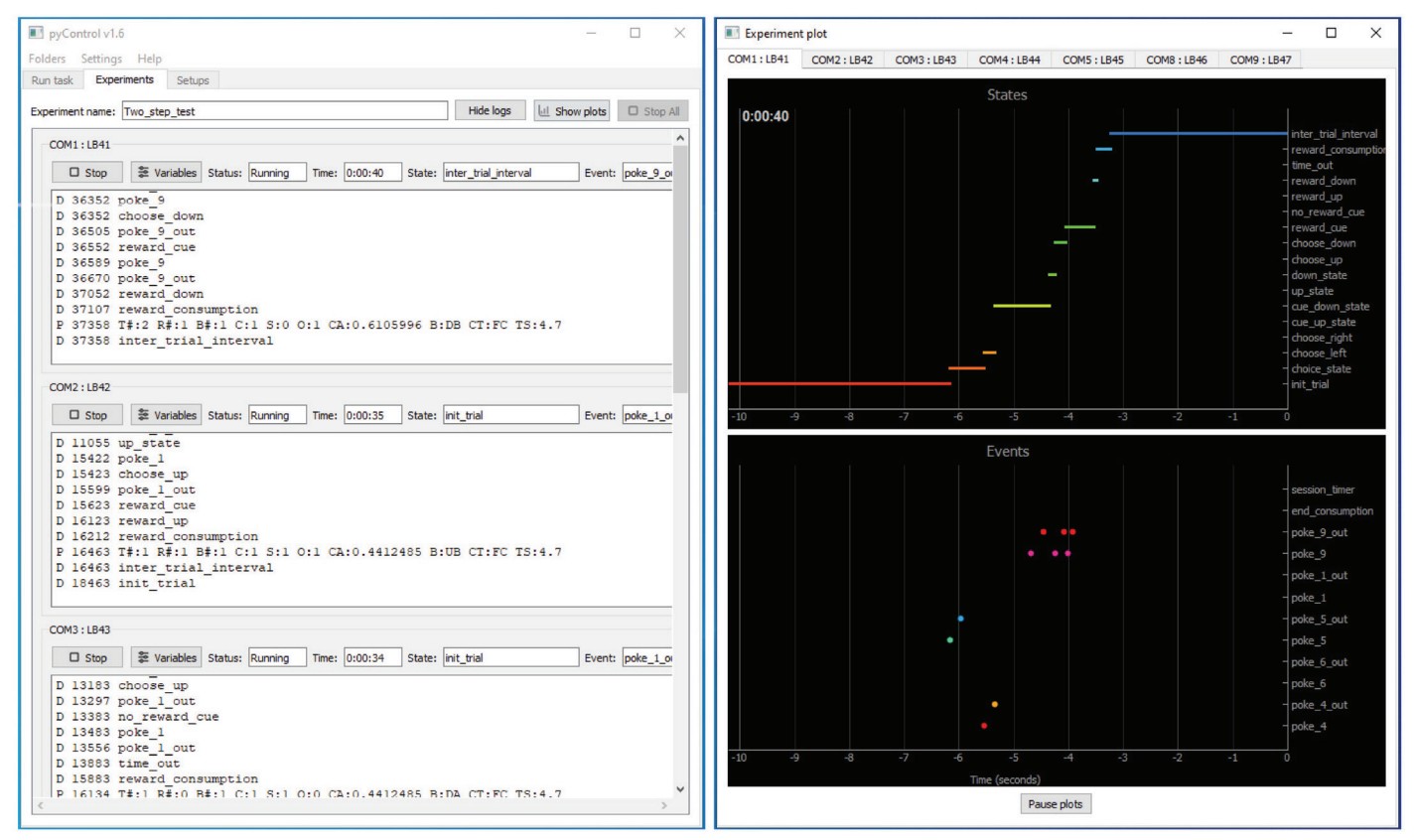

**Figure 4.** pyControl graphical user interface (GUI). The GUI's *Experiments* tab is shown on the left running a multi-subject experiment, with the experiment's plot window open on the right showing the recent states and events for one subject. For images of the other GUI functionality, see the GUI docs.

A tabbed plot window can be opened showing live scrolling plots of the events, states, and analog signals for each subject, and individual subjects' plots can be undocked to allow behaviour of multiple subjects to be visualised simultaneously.

The GUI is implemented entirely in Python using the PyQt GUI framework and PyQtGraph plotting library. The GUI is cross-platform and has been used on Windows, Mac, and Linux, though most development and testing has been under Windows. The code is organised into modules for communication with the pyboard, different GUI components, and data visualisation.

## pyControl data

Data from pyControl sessions are saved as text files (see *Figure 1—figure supplement 1* for an example). When a session starts, information including the subject, task and experiment names, and start data and time, are written to the data file. While the task is running, all events and state transitions are saved automatically with millisecond timestamps. The user can output additional data by using the *print* function in their task file. This outputs the printed line to the computer, where it is displayed in the log and saved to the data file, along with a timestamp. In decision-making tasks, we typically print one line each trial indicating the trial number, the subject's choice, and trial outcome, along with any other relevant task variables. If an error occurs while the framework is running, a traceback reporting the error and line number in the task file where it occurred is displayed in the log and written to the data file. Continuous data from analog inputs is saved in separate binary files.

In addition to data files, task definition files used to generate data are copied to the experiment's data folder, with a file hash appended to the file name that is also recorded in the corresponding session's data file. This ensures that every task file version used in an experiment is automatically saved with the data, and it is always possible to uniquely identify the specific task file used for a particular session. If any variables are changed from default values in the task file, this is automatically recorded

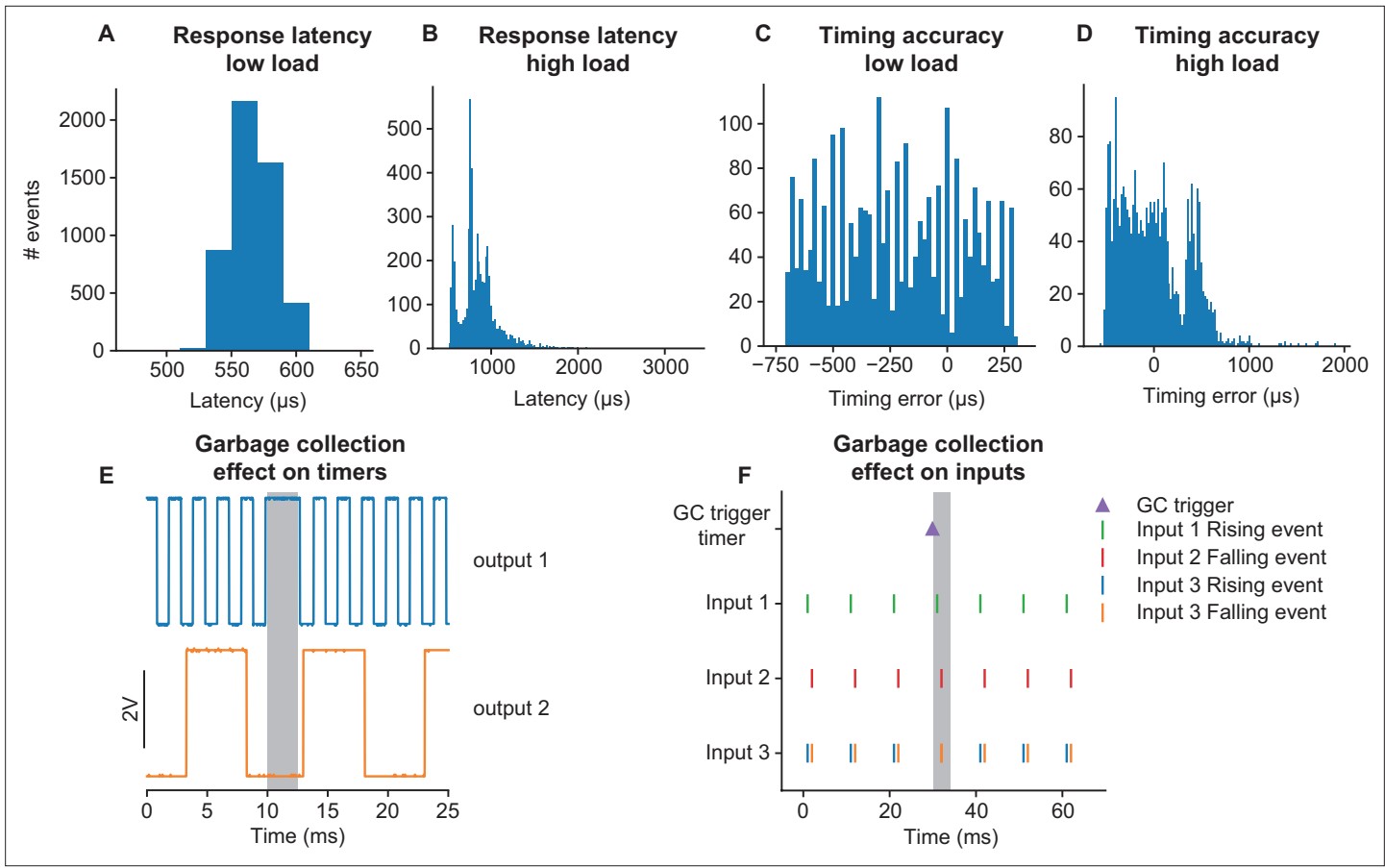

**Figure 5.** Framework performance. (**A**) Distribution of latencies for the pyControl framework to respond to a change in a digital input by changing the level of a digital output. (**B**) As (**A**) but under a high load condition (see main text). (**C**) Distribution of pulse duration errors when framework generates a 10 ms pulse. (**D**) As (**C**) but under a high load condition. (**E**) Effect of MicroPython garbage collection on pyControl timers. Signals are two digital outputs, one toggled on and off every 1 ms (blue), and one every 5 ms (orange), using pyControl timers. The 1 ms timer that that elapsed during garbage collection (indicated by grey shading) was processed once garbage collection had finished, causing a short delay. Garbage collection had no effect on the 5 ms timer that was running but did not elapse during garbage collection. (**F**) Effect of garbage collection on pyControl inputs. A signal comprising 1 ms pulses every 10 ms was received by three pyControl digital inputs. Input 1 was configured to generated framework events on rising edges (green), input 2 on falling edges (red), and input 3 on both rising (blue) and falling (orange) edges. Garbage collection (indicated by grey shading) was triggered 1 ms before an input pulse. Inputs 1 and 2 both generated their event that occurred during garbage collection with the correct timestamp. If multiple events occur on a single digital input during a single garbage collection, only the last event is generated correctly, causing the missing rising event on input 3.

in the session's data file. These automatic self-documenting features are designed to promote replicability of pyControl experiments. We encourage users to treat the versioned task files as part of the experiment's data and include them in data repositories.

Modules are provided for importing data files into Python for analysis and for visualising sessions offline. Importing a data file creates a Session object with attributes containing the session's information and data. For convenience, two representations of the state and event data are generated: (1) a dictionary whose keys are event and state names, and values are NumPy arrays with the corresponding event or state entry times, and (2) a list of events and state entries in the order they occurred, whose elements are named tuples with the event/state name and timestamp as attributes. For more information, see the data docs.

## Framework performance

To validate the performance of the pyControl framework, we measured the system's response latency and timing accuracy. Response latency was assessed using a task which set a digital output to match the state of a digital input driven by a square wave signal. We recorded the input and output signals

and plot the distribution of latencies between the two signals across all rising and falling edges (*Figure 5A and B*). In a 'low load' condition where the pyboard was not processing other inputs, response latency was 556 ± 17 μs (mean ± SD). This latency reflects the time to detect the change in the input, trigger a state transition, and update the output during processing of the 'entry' event in the new state. We also measured response latency in a 'high load' condition where the pyboard was additionally monitoring two digital inputs each generating framework events in response to edges occurring as Poisson processes with an average rate of 200 Hz, and acquiring signal from two analog inputs at 1 kHz sample rate each. In this high load condition, the response latency was 859 ± 241 μs (mean ± SD), the longest latency recorded was 3.3 ms with 99.6% of latencies < 2 ms.

To assess timing accuracy, we used a task which turned on a digital output for 10 ms when a rising edge was received on a digital input. The input was driven by a 51 Hz square wave to ensure that the timing of input edges drifted relative to the framework's 1 ms clock ticks. We plot the distribution of errors between the measured durations of the output pulses and the 10 ms target duration (*Figure 5C and D*). In the low load condition, timing errors were approximately uniformly distributed across 1 ms (mean error –220 μs, SD 282 μs), as expected given the 1 ms resolution of the pyControl framework clock ticks. In the high load condition, timing variability was only slightly increased (mean –10 μs, SD 353 μs), with the largest recorded error 1.9 ms and 99.5% of errors < 1 ms. Overall, these data show that the framework's latency and timing accuracy are sufficient for the great majority of neuroscience applications, even when operating under loads substantially higher than experienced in typical tasks.

Users who require very tight timing/latency performance should be aware of MicroPython's automatic garbage collection. Garbage collection is triggered when needed to free up memory and takes a couple of milliseconds. Normal code execution is paused during garbage collection, though interrupts (used to register external inputs and update the framework clock) run as normal. pyControl timers that elapse during garbage collection are processed once it has completed (*Figure 5E*). Timers that are running but do not elapse during garbage collection are unaffected. Digital inputs that occur during garbage collection are registered with the correct timestamp (*Figure 5F*), but will only be processed once garbage collection has completed. The only situation where events may be missed due to garbage collection is if a single digital input receives multiple event-triggering edges during a single garbage collection, in which case only the last event is processed correctly (*Figure 5F*). To avoid garbage collection affecting critical processing, the user can manually trigger garbage collection at a time when it will not cause problems (see MicroPython docs), for example, during the inter-trial interval (ITI). In the latency and timing accuracy validation experiments (*Figure 5A–D*), garbage collection was triggered by the task code at a point in the task where it did not affect the measurements.

A final constraint is that as each event takes time to process, there is a maximum *continuous* event rate above which the framework cannot process events as fast as they occur, causing the event queue to grow until available memory is exhausted. This rate will depend on the processing triggered by each event, but is approximately 960 Hz for digital inputs triggering state transitions but no additional processing. In practice, we have never encountered this when running behavioural tasks as average event rates are typically orders of magnitude lower and transiently higher rates are buffered by the queue.

## Application examples

We illustrate how pyControl is used in practice with example applications in operant box, head-fixed, and maze-based tasks. Task and hardware definition files for these experiments are provided in the article's data repository. For additional use cases, see also *Korn et al., 2021*; *Akam et al., 2021*; *Koralek and Costa, 2020*; *Nelson et al., 2020*; *Blanco-Pozo et al., 2021*; *van der Veen et al., 2021*; *de Barros et al., 2021*; *Samborska et al., 2021*; *Kilonzo et al., 2021*; *Strahnen et al., 2021*.

## 5-choice serial reaction time task (5-CSRT)

The 5-CSRT is a long-standing and widely used assay for measuring sustained visual attention and motor impulsivity in rodents (*Carli et al., 1983*; *Bari et al., 2008*). The subject must detect a brief flash of light presented pseudorandomly in one of five nose-poke ports and report the stimulus location by poking the port to trigger a reward delivered to a receptacle on the opposite wall.

We developed a custom operant box for the 5-CSRT (*Figure 6A and B*), discussed in detail in a separate manuscript (*Kapanaiah et al., 2021*). The pyControl hardware comprised a breakout board connected to a 5-poke board, which integrates the IR beams and stimulus LEDs for the 5-choice ports

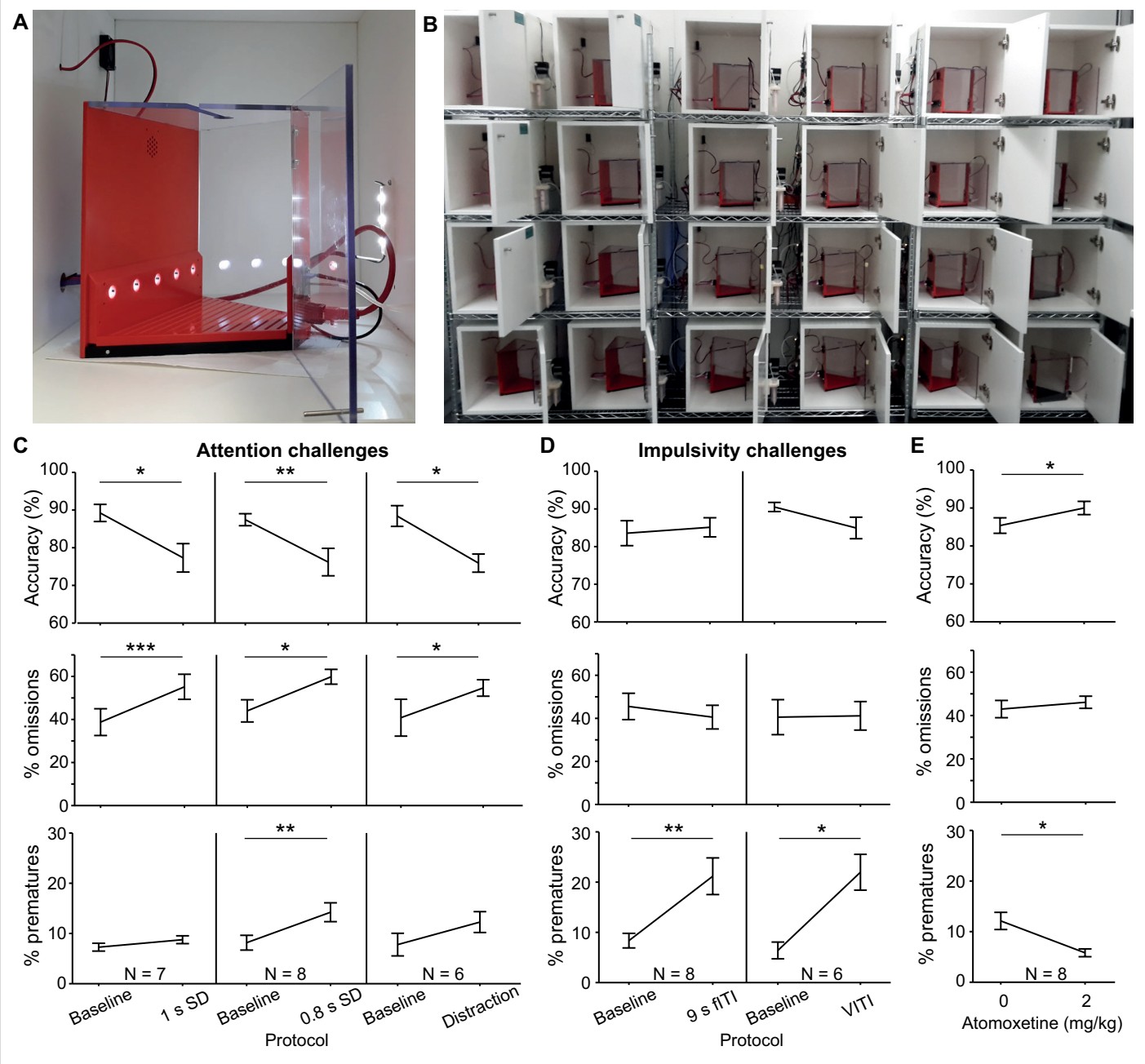

**Figure 6.** 5-choice serial reaction time task (5-CSRTT). (**A**) Trapezoidal operant box with 5-choice wall (poke-holes shown illuminated) within a sound-attenuated cubicle. (**B**) High-throughput training setup comprising 24 operant boxes. (**C, D**) Performance measures on the 5-CSRTT during protocols challenging either sustained attention – by shortening the SD or delivering a sound distraction during the waiting time (**C**) or motor impulsivity – by extending the inter-trial interval (ITI) to a fixed (fITI) or variable (vITI) length (**D**). Protocols used are indicated by x-axes. Note the rather selective decrease of attentional performance (accuracy, %omissions) or impulse control (%prematures) achieved by the respective challenges. (**E**) Validation of the possibility to detect cognitive enhancement in the 5-CSRTT (9s-fITI challenge) by application of atomoxetine, which increased attentional accuracy and decreased premature responding, as predicted. Asterisks in (**C–E**) indicate significant within-subject comparisons relative to the baseline (2 s SD, 5 s fITI; **C, D**) or the vehicle (**E**) condition (paired-samples $t$-test). *$p<0.05$, *$p<0.01$, *$p<0.001$. Error bars display s.e.m. Note that two mice of the full cohort (N = 8) did not participate in all challenges as they required more training time to reach the baseline stage.

The online version of this article includes the following figure supplement(s) for figure 6:

**Figure supplement 1.** Hardware configuration for 5-choice serial reaction time task (5-CSRTT).

on a single PCB, a single poke board for the reward receptacle, an audio board, and a stepper motor board to control a peristaltic pump for reward delivery (*Figure 6—figure supplement 1*).

To validate the setup, a cohort of eight C57BL/6 mice was trained in the 5-CSRTT using a staged training procedure (see Materials and methods). The baseline protocol reached at the end of training used a stimulus duration (SD) of 2 s and a 5 s ITI from the end of reward consumption to the presentation of the next stimulus. These task parameters were then manipulated to challenge subject's ability to either maintain sustained attention or withhold impulsive premature responses. Attention was challenged in three conditions: by decreasing the SD to either 1 s or 0.8 s, or by an auditory distraction of 70 dB white noise, played between 0.5 s and 4.5 s of the 5 s ITI. In all three attention challenges, the accuracy with which subjects selected the correct port – the primary measure of sustained attention – decreased (p<0.05; paired *t*-tests comparing accuracy under the prior baseline protocol to accuracy under the challenge condition, *Figure 6C*). Also, as expected, omissions (i.e. failures to poke any port in the response window) increased (p<0.05, *t*-test). In the attention challenges, the rate of premature responses – the primary measure of impulsivity – remained either unchanged (1 s SD challenge, auditory distraction; p>0.1, *t*-test) or changed to a comparatively small extent (0.8 s SD challenge, p<0.01, *t*-test). Similarly, when impulsivity was challenged by extending the ITI, to either a 9 s fixed ITI (fITI) or to a pseudo-randomly varied ITI length (vITI), premature responses increased strongly (p<0.05, *t*-test), while attentional accuracy and omissions did not (*Figure 6D*). This specificity of effects of the challenges was as good – if not better – than that achieved by us previously in a commercial set-up (Med Associates, Inc; *Grimm et al., 2018*).

We further validated the task implementation by replicating effects of a pharmacological treatment – atomoxetine – that has been shown to reduce impulsivity in the 5-CSRTT (*Navarra et al., 2008*; *Paterson et al., 2011*). Using the 9 s fITI impulsivity challenge, we found that 2 mg/kg atomoxetine could reliably reduce premature responding and increase attentional accuracy (p<0.05, paired *t*-test comparing performance under vehicle vs. atomoxetine; *Figure 6E*), consistent with its previously described effect in this rodent task (*Navarra et al., 2008*; *Paterson et al., 2011*; *Pillidge et al., 2014*; *Fitzpatrick and Andreasen, 2019*).

## Vibrissae-based object localisation task

We illustrate pyControl's utility for head-fixed behaviours with a version of the vibrissae-based object localisation task (*O'Connor et al., 2010*). Head-fixed mice used their vibrissae (whiskers) to discriminate the position of a pole moved into the whisker field at one of two different anterior-posterior locations (*Figure 7A*). The anterior 'Go' location indicated that licking in a response window after stimulus presentation would deliver a water reward, while the posterior 'NoGo' location indicated that licking in the response window would trigger a timeout (*Figure 7B*). Unlike in the original task, mice were positioned on a treadmill allowing them to run. Although running was not required to perform the task, we observed 10–20 s running bouts alternated with longer stationary periods (*Figure 7C*), in line with previous reports (*Ayaz et al., 2019*). pyControl hardware used to implement the setup comprised a breakout board, a stepper motor driver to control the anterior-posterior position of the stimulus, a lickometer, and a rotary encoder to measure running speed (*Figure 7—figure supplement 1*).

Mice were first familiarised with the experimental setup by head-fixing them on the treadmill for increasingly long periods of time (5–20 min) over 3 days. From the fourth day, mice underwent a 'detection training', during which the pole was only presented in the Go position, and water automatically delivered after each stimulus presentation. We then progressively introduced NoGo trials and made water delivery contingent on the detection of one or more licks in the response window. Subjects reached 75% correct performance within 5–9 days from the first training session, at which point, they were trained for at least three further days to make sure that they had reliably learned the task (*Figure 7D*). Early in training, mice frequently licked prior to and during stimulus presentation, as well as during the response window, on both Go and NoGo trials (*Figure 7E*). Following learning, licking prior to and during stimulus presentation was greatly reduced, and mice licked robustly during the response window on Go trials and withheld licking on NoGo trials, performing a high percentage of hit and correct rejection trials (*Figure 7F*).

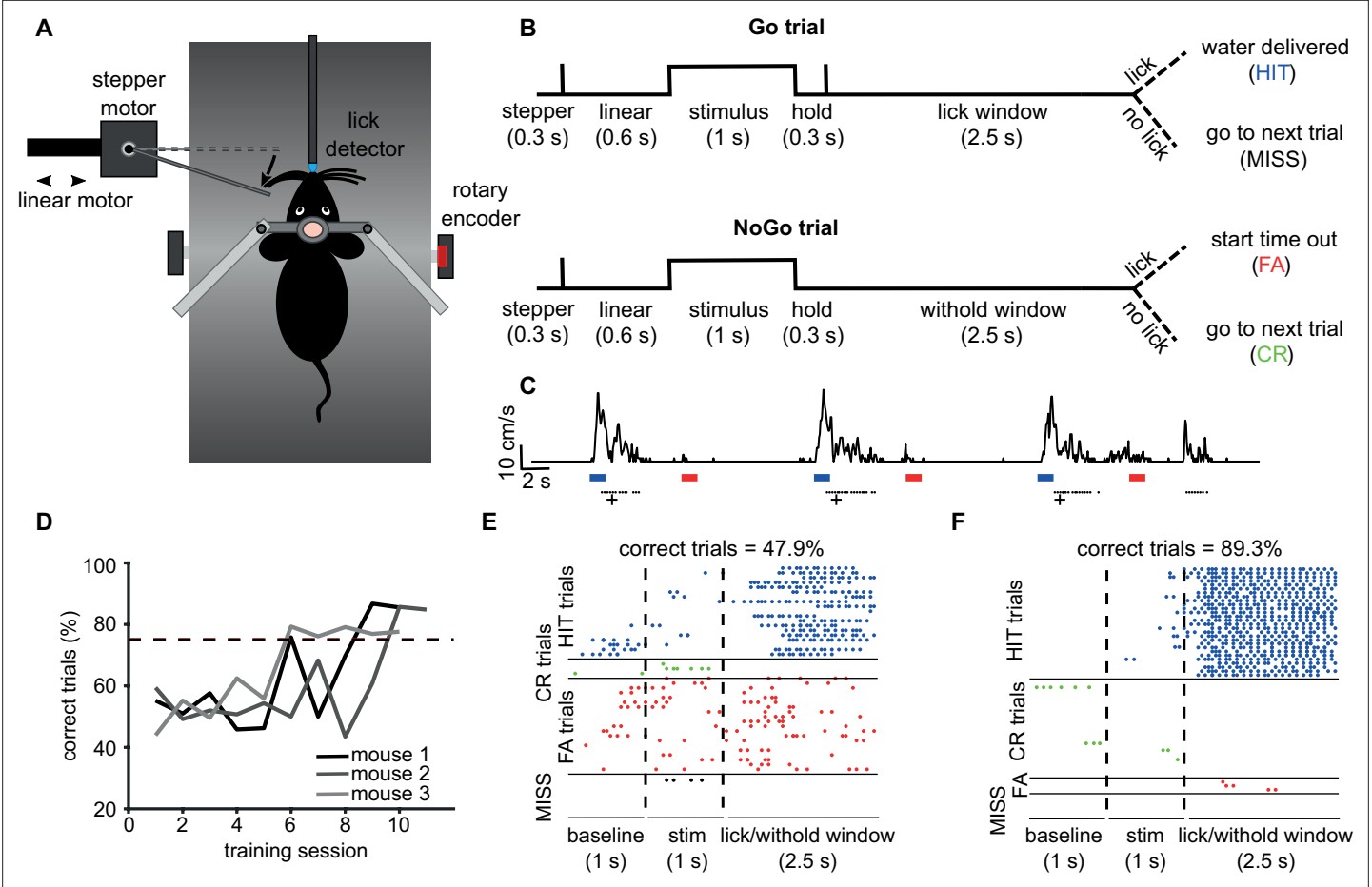

**Figure 7.** Vibrissae-based object localisation task. (**A**) Diagram of the behavioural setup. Head-fixed mice were positioned on a treadmill with their running speed monitored by a rotary encoder. A pole was moved into the whisker field by a linear motor, with the anterior-posterior location controlled using a stepper motor. Water rewards were delivered via a spout positioned in front of the animal and licks to the spout were detected using an electrical lickometer. (**B**) Trial structure: before stimulus presentation, the stepper motor moved into the trial position (anterior or posterior). Next, the linear motor translated the stepper motor and the attached pole close to the mouse's whisker pad, starting the stimulation period. A lick window (during Go trials) or withhold window (during NoGo trials) started after the pole was withdrawn. FA, false alarm; CR, correct rejection. (**C**) pyControl simultaneously recorded running speed (top trace) and licks (black dots) of the animals, as well as controlling stimulus presentation (blue and red bars for Go and NoGo stimuli) and solenoid opening (black crosses). (**D**) Percentage of correct trials for three mice over the training period. Mice were considered expert on the task after reaching 75% correct trials (dotted line) and maintaining such performance for three consecutive days. (**E**) Detected licks before, during, and after tactile stimulation, during an early session before the mouse has learned the task, sorted by trial type: hit trials (blue), correct rejection trials (green), false alarm trials (red), and miss trials (black). Each row is a trial, each dot is a detected lick. Correct trials for this session were 47.9% of total trials. (**F**) As (**E**) but for data from the same mouse after reaching the learning threshold (correct trials = 89.3% of total trials).

The online version of this article includes the following figure supplement(s) for figure 7:

**Figure supplement 1.** Hardware configuration for vibrissae-based object localisation task.

## Social decision-making task

Our final application example is a maze-based social decision-making task for mice, adapted from that developed for rats by *Márquez et al., 2015*. In this task, a 'focal' animal's choices determine reward delivery for a 'recipient' animal, allowing preference for 'prosocial' vs. 'selfish' choices to be examined. The behavioural apparatus comprised an automated double T-maze (*Figure 8—figure supplement 1*). Each T-maze consisted of a central corridor with nose-poke ports on each side (choice area) and two side arms each with a food receptacle connected to a pellet dispenser at the end (*Figure 8A and B*). Access from the central choice area to the side arms was controlled by pneumatic doors.

The task comprised two separate stages: (1) individual training, where animals learn to open doors by poking the ports in the central arms and retrieve pellets in the side arms; and (2) social testing,

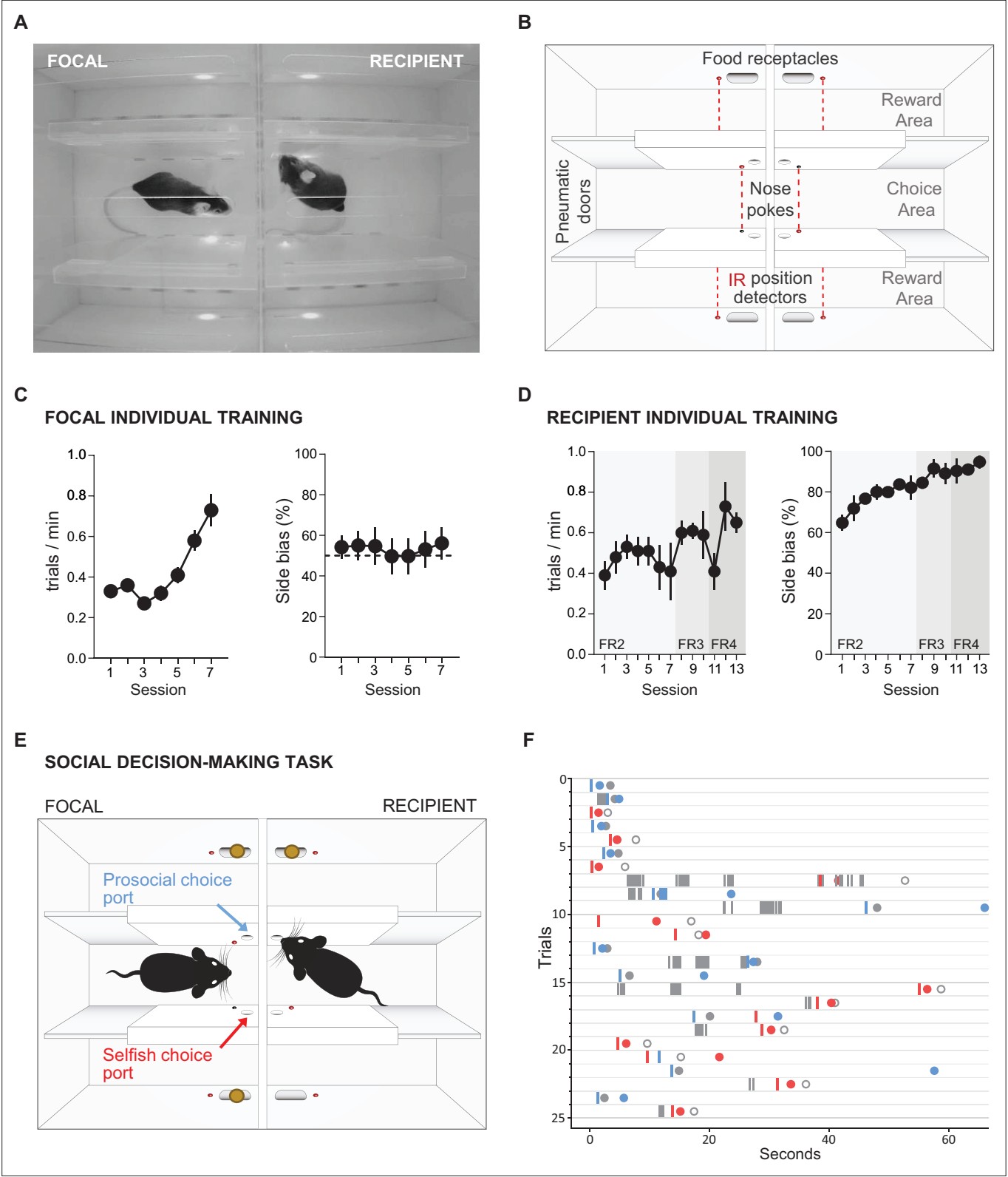

**Figure 8.** Social decision-making task. (**A**) Top view of double T-maze apparatus showing two animals interacting during social decision-making. (**B**) Setup diagram; in each T-maze, nose-pokes are positioned on either side of the central choice area. Sliding pneumatic doors give access to the side arms of each maze (top and bottom in diagram) where pellet dispensers deliver food rewards. Six IR beams (depicted as grey and red circles connected by a dotted red line) detect the position of the animals to safely close the doors once access to an arm is secured. (**C**) Focal animal individual training

*Figure 8 continued on next page*

*Figure 8 continued*

showing the number of trials completed per minute (left panel) and side bias (right panel) across days of training. (**D**) As (**C**) but for the recipient animal. (**E**) Social decision-making task. The trial starts with both animals in the central arm. The recipient animal has learnt in previous individual training to poke the port on the upper side of the diagram to give access to a food pellet in the corresponding reward area. During the social task, the recipient animal's ports no longer control the doors but the animal can display food-seeking behaviour by repeatedly poking the previously trained port. The focal animal has previously learned in individual training to collect food from the reward areas on both sides (top and bottom of diagram) by poking the corresponding port in the central choice area to activate the doors. During social decision-making, the focal animal can either choose the 'prosocial' port, giving both animals access to the side (upper on diagram) of their respective mazes where both receive reward, or can choose the 'selfish' port, giving both animals access to the other side (lower on diagram) where only the focal animal receives reward. (**F**) Raster plot showing behaviour of a pair of animals over one session during early social testing. Nose-pokes are represented by vertical lines, and colour coded according to the role of each mouse and choice type (grey, recipient's pokes, which are always directed towards the prosocial side; blue, focal's pokes in the prosocial choice port; red, focal's pokes in selfish port). Note that latency for focal choice varies depending on the trial, allowing the recipient to display its food-seeking behaviour or not. Circles indicate the moment where each animal visits the food-receptacle in their reward arm. Focal animals are always rewarded, and the colour of the filled circle indicates the type of trial after decision (blue, prosocial choice; red, selfish choice). Grey circles indicate time of receptacle visit for recipients, where filled circles correspond to prosocial trials, where recipient is also rewarded, and open circles to selfish trials, where no pellet is delivered.

The online version of this article includes the following figure supplement(s) for figure 8:

**Figure supplement 1.** Hardware configuration for social decision-making task.

where the decisions of the focal animal control the doors in both mazes, and hence determine rewards for both itself and the recipient animal in the other maze.

The individual training protocols were different for the focal and recipient animals. During individual training for the focal animal, a single poke in either port in the central arm opened the corresponding door, allowing access to a side arm. Accessing either side arm was rewarded with a pellet at the food receptacle in the arm. Under this schedule, subjects increased their rate of completing trials over seven training days (*Figure 8C*, repeated measures ANOVA $F_{(6,42)} = 12.566$, p=0.000004) without developing a bias for either side of the maze (p>0.27 for all animals, *t*-test). During individual training for the recipient animal, only one of the nose-poke ports in the central arm was active, and the number of pokes required to open the corresponding door increased over 13 days of training, with four pokes eventually required to access the side arm to obtain a pellet in the food receptacle. Under this schedule, the recipient animals developed a strong preference for the active poke over the course of training (*Figure 8D*, right panel, repeated measures ANOVA $F_{(12,24)} = 3.908$, p=0.002), with approximately 95% of pokes directed to the active side by the end of training.

During social testing, the two animals were placed in the double T-maze, one in each T, separated by a transparent perforated partition that allowed the animals to interact using all sensory modalities. The doors in the recipient animal's maze were no longer controlled by the recipient animal's pokes, but were rather yoked to the doors of the focal animal, such that a single poke to either port in the focal animals choice area opened the doors in both mazes on the corresponding side. As in individual training, the focal animal was rewarded for accessing either side, while the recipient animal was rewarded only when it accessed one side of the maze. The choice made by the focal animal therefore determined whether the recipient animal received reward, so the focal animal could either make 'prosocial' choices which rewarded both it and the recipient, or 'selfish' choices which rewarded only the focal animal. As a proof of concept, we show nose-pokes and reward deliveries from a pair of interacting mice from one social session (*Figure 8F*). A full analysis of the social behaviour in this task will be published separately (Esteve-Agraz and Marquez, in preparation).

## Discussion

pyControl is an open-source system for running behavioural experiments, whose principal strengths are (1) a flexible and intuitive Python-based syntax for programming tasks; (2) inexpensive, simple, and extensible behavioural hardware that can be purchased commercially or assembled by the user; (3) a GUI designed for efficiently running high-throughput experiments on many setups in parallel from a single computer; and (4) extensive online documentation and user support.

pyControl can contribute to behavioural neuroscience in two important ways: first, it makes it quicker, easier, and cheaper to implement a wide range of behavioural tasks and run them at scale. Second, it facilitates communication and reproducibility of behavioural experiments, both because

the task definition syntax is highly readable and because self-documenting features ensure that the exact task version and parameters used to generate data are automatically stored with the data itself.

pyControl's strengths and limitations stem from underlying design choices. We will discuss these primarily in relation to two widely used open-source systems for experiment control in neuroscience Bpod (Josh Sanders) and Bonsai (*Lopes et al., 2015*). Bpod is a useful point of comparison as it is probably the most similar project to pyControl in terms of functionality and implementation, Bonsai because it represents a very different but powerful formalism for controlling experiments that is often complementary. Space constraints preclude detailed comparison with other projects, but see *Devarakonda et al., 2016*; *O'Leary et al., 2018*; *Kim et al., 2019*; *Gurley, 2019*; *Saunders and Wehr, 2019*; *Bhagat et al., 2020*; *Buscher et al., 2020*.

Both pyControl and Bpod provide a state machine-based task definition syntax in a high-level programming language, run the state machine on a microcontroller, have commercially available open-source hardware, graphical interfaces for controlling experiments, and are reasonably mature systems with a substantial user base beyond the original developers. Despite these commonalities, there are significant differences which are useful for prospective users to understand.

The first is that in pyControl user-created task definition code runs directly on a pyboard microcontroller, supported by framework code that determines when user-defined functions are called. This contrasts with Bpod, where user code written in either MATLAB (Bpod) or Python (PyBpod) is translated into instructions passed to the microcontroller, which itself runs firmware implemented in the lower-level language C++. These two approaches offer distinct advantages and disadvantages.

Running user Python code directly on the microcontroller avoids separating the task logic into two conceptually distinct levels – flexible code written in a high-level language that runs on the computer, and the more constrained set of operations supported by the microcontroller firmware. Our understanding of how this works in Bpod is that the high-level user code implements a loop over trials where each loop defines a finite state machine for the current trial – specifying for each state which outputs are on and which events trigger transitions to which other states, then uploads this information to the microcontroller, runs the state machine until it reaches an exit condition indicating the end of the trial, and finally receives information from the microcontroller about what happened before starting the next trial's loop. The microcontroller firmware implements some functionality beyond a strict finite state machine formalism, including timers and event counters that are not tied to a particular state, but does not support arbitrary user code or variables. We suggest readers consult the relevant documentation (pyControl, Bpod, PyBpod) and example tasks (pyControl, Bpod, PyBpod) to compare syntaxes directly. A second advantage of running user code directly on the microcontroller is that the user has direct access from their task code to microcontroller functionality such as serial communication. A third is that the pyControl framework (as well as the GUI) is written in Python rather than C++, facilitating code maintenance, and lowering the barrier to users extending system functionality.

The two principal disadvantages of running the task entirely on the microcontroller are (1) although modern microcontrollers are very capable, their resources are more limited than a computer – which constrains how computationally and memory-intensive task code can be and precludes using modules such as NumPy. (2) Lack of access to the computer from task code, for example, to interact with other programs or display custom plots. To address these limitations, we are currently developing an application programming interface (API) to allow pyControl tasks running on the microcontroller to interact with user code running on the computer. This will work via the user defining a Python class with methods that get called at the start and end of the run for initial setup and post-run clean-up, as well as an update method called regularly during the run with any new data received from the board as an argument.

There are also differences in hardware design. The two most significant are (1) the pyControl breakout board tries to make connectors (behaviour ports and BNC) as flexible as possible at the cost of not being specialised for particular functions. Bpod tends to use a given connector for a specific function; for example, it has separate *behaviour ports* and *module ports*, with the former designed for controlling a nose-poke, and the latter for UART serial communication with external modules. Practically, this means that pyControl exposes microcontroller pins (which often support multiple functions)

directly on connectors whereas Bpod tends to incorporate intervening circuitry such as electrical isolation for BNC connectors and serial line driver ICs on module ports. (2) Bpod uses external modules, each with its own microcontroller and C++ firmware, for functions which pyControl implements using the microcontroller on the breakout board, specifically analog input and output, I2C serial communication, and acquiring signal from a rotary encoder. These design choices make pyControl hardware simpler and cheaper. Purchased commercially the Bpod state machine costs $765 compared to €250 for the pyControl breakout board, and Bpod external modules each cost hundreds of dollars. This is not to say that pyControl necessarily represents better value; a given Bpod module may offer more functionality (e.g. more channels, higher sample rates). But the two systems do represent different design approaches.

Both the pyControl and PyBpod GUIs support configuring and running experiments on multiple setups in parallel from a single computer, while the MATLAB-based Bpod GUI controls a single setup at a time. Their user interfaces are each very different; the respective user guides (pyControl, Bpod, PyBpod) give the best sense for the different approaches. We think it is a strength of the pyControl GUI, reflecting the relative simplicity of the underlying code base, that scientist users not originally involved in the development effort have made substantial contributions to its functionality (see GitHub pull requests).

Bonsai (*Lopes et al., 2015*) represents a very different formalism for experiment control that is not based around state machines. Instead, the Bonsai user designs a *dataflow* by arranging and connecting nodes in a graphical interface, where nodes may represent data sources, processing steps, or outputs. Bonsai can work with a diverse range of data types including video, audio, analog, and digital signals. Multiple data streams can be processed in parallel and combined via a rich set of operators including arbitrary user code. Bonsai is very powerful, and it is likely that any task implemented in pyControl could also be implemented in Bonsai. The reverse is certainly not true as Bonsai can perform computationally demanding real-time processing on high-dimensional data such as video, which is not supported by pyControl.

Nonetheless, in applications where either system could be used, there are reasons why prospective users might consider pyControl: (1) pyControl's task definition syntax may be more intuitive for tasks where (extended) state machines are a natural formalism. The reverse is true for tasks requiring parallel processing of multiple complex data streams. (2) pyControl is explicitly designed for efficiently running high-throughput experiments on many setups in parallel. Though it is possible to control multiple hardware setups from a single Bonsai dataflow, Bonsai does not explicitly implement the concept of a multi-setup experiment so the user must duplicate dataflow components for each setup themselves. As task parameters and data file names are specified across multiple nodes in the dataflow, configuring these for a cohort of subjects can be laborious – though it is possible to automate this by calling Bonsai's command line interface from user-created Python scripts. (3) pyControl hardware modules can simplify the physical construction of behavioural setups. Though Bonsai itself is software, some compatible behavioural hardware has been developed by the Champalimaud Foundation Hardware Platform (https://www.cf-hw.org/harp), which offers tight timing synchronisation and close integration with Bonsai, though documentation is currently limited. In practice, we think the two systems are often complementary; for example, we use Bonsai in our workflow for acquiring and compressing video data from sets of pyControl operant boxes (GitHub; *Akam, 2020*), and we hope to integrate them more closely in future. pyControl is under active development. We are currently prototyping a home cage training system which integrates a pyControl operant box with a mouse home cage via an access control module which allows socially housed animals to individually access the operant box to train themselves with minimal user intervention. We are also developing hardware to enable much larger-scale behavioural setups, such as complex maze environments with up to 68 behaviour ports per setup. As discussed above, we are finalising an API to allow pyControl tasks to interact with user Python code running on the computer.

In summary, pyControl is a user-friendly and flexible tool addressing a commonly encountered use case in behavioural neuroscience; defining behavioural tasks as extended state machines, running them efficiently as high-throughput experiments, and communicating task logic to other researchers.

# Materials and methods

**Key resources table**

| Reagent type (species) or resource | Designation | Source or reference | Identifiers | Additional information |
|---|---|---|---|---|
| Software, algorithm | pyControl | https://github.com/pyControl/code | RRID:SCR_021612 | Repository containing pyControl GUI and framework code |
| Other | pyControl hardware | https://github.com/pyControl/hardware | RRID:SCR_021612 | Repository containing pyControl hardware designs |
| Other | pyControl Docs | https://pycontrol.readthedocs.io; a PDF version of the docs is included in supplementary material | RRID:SCR_021612 | pyControl documentation |

pyControl task files used in all experiments, and data and analysis code for the performance validation experiments, are included in the article's data and code repository.

## Framework performance validation

Framework performance was characterised using pyboards running MicroPython version 1.13 and pyControl version 1.6. Electrical signals used to characterise response latency and timing accuracy (*Figure 5*) were recorded at 50 kHz using a PicoScope 2204A USB oscilloscope.

To assess response latency (*Figure 5A and B*), a pyboard running the task file *input_follower.py* received a 51 Hz square wave input generated by the PicoScope's waveform generator. The task turned an output on and off to match the state of the input signal. The latency distribution was assessed by recording 50 s of the input and output signals and evaluating the latency between the signals at each rising and falling edge.

To assess timing accuracy (*Figure 5C and D*), a pyboard running the task file *triggered_pulses.py* received a 51 Hz square wave input generated by the PicoScope's waveform generator. The task triggered a 10 ms output pulse whenever a rising edge occurred in the input signal. The output signals were recorded for 50 s, and the duration of each output pulses was measured to assess the distribution of timing errors.

In both cases, the experiments were performed separately in a low load and high load condition. In the low load condition, the task was not monitoring any other inputs. In the high load condition, the task was additionally acquiring data from two analog inputs at 1 kHz sample rate each, and monitoring two digital inputs, each of which was generating framework events in response to edges occurring as a Poisson process with average rate 200 Hz. These Poisson input signals were generated by a second pyboard running the task *poisson_generator.py*.

To assess the effect of garbage collection on pyControl timers (*Figure 5E*), the task file *gc_timer_test.py* was run on a pyboard. This uses pyControl timers to toggle one digital output on and off every 1 ms and another every 5 ms. The resulting signals were recorded using the PicoScope and plotted around a garbage collection episode identified by visually inspecting the 1 ms timer signal.

To assess the effect of garbage collection on digital input processing (*Figure 5F*), a signal comprising 1 ms pulses every 10 ms was generated using the PicoScope, and connected to three digital inputs on a pyboard running the task *gc_inputs_test.py*. The task configures one input to generate events on rising edges, one on falling edges, and one on both rising and falling edges, and uses a pyControl timer to trigger garbage collection 1ms before a subset of the input pulses. Event times recorded by pyControl were plotted to generate the figure.

Analysis and plotting of the framework validation data was performed in Python using code included in the data repository.

## Application examples

The 5-CSRTT5.

## Animals

The 5-CSRTT experiment used a cohort of eight male C57BL/6 mice, aged 3–4 months at the beginning of training. Animals were group-housed (2–3 mice per cage) in Type II-Long individually ventilated cages (Greenline, Tecniplast, G), enriched with sawdust, sizzle-nest, and cardboard houses (Datesand, UK), and subjected to a 13 hr light/11 hr dark cycle. Mice were kept under food restriction at 85–95% of their average free-feeding weight which was measured over 3 days immediately prior to the start of food restriction at the start of the behavioural training. Water was available ad libitum.

This experiment was performed in accordance to the German Animal Rights Law (Tierschutzgesetz) 2013 and approved by the Federal Ethical Review Committee (Regierungsprädsidium Tübingen) of Baden-Württemberg.

## Behavioural hardware

The design of the operant boxes for the 5-CSRTT setups is discussed in detail in a separate manuscript (*Kapanaiah et al., 2021*). Briefly, the box had a trapezoidal floorplan with the 5-choice wall at the wide end and reward receptacle at the narrow end of the trapezoid to minimise the floor area and hence reduce distractions. The side walls and roof were made of transparent acrylic to allow observation of the animal, the remaining walls were made from opaque PVC to minimise visual distractions (*Figure 6A*). Design files for the operant box, and peristaltic and syringe pumps for reward delivery, are at https://github.com/KaetzelLab/Operant-Box-Design-Files; *Kaetzell, 2021*. Potentially distracting features (house light, cables) were located outside of the box and largely invisible from the inside. The pyControl hardware used and the associated hardware definition are shown in *Figure 6—figure supplement 1*. The operant box was enclosed by a sound attenuating chamber, custom made in 20 mm melamine-coated MDF, adapted from a design in the hardware repository. The pyControl breakout boards, and other PCBs that were not integrated into the box itself, were mounted on the

**Table 2.** 5-choice serial reaction time task (5-CSRTT) training and challenge stages.
The parameters stimulus duration (SD) and inter-trial interval (ITI, waiting time before stimulus) are listed for each of the five training stages (S1–5) and the subsequent challenge protocols on which performance was tested for 1 day each (C1–5). For the training stages, performance criteria which had to be met by an animal on two consecutive days to move to the next stage are listed on the right. See Materials and methods for the definition of these performance parameters.

**5-CSRTT training**

| Stage | Task parameters | | Criteria for stage transition (two consecutive days) | | | |
| | SD (s) | ITI (s) | # correct | % correct | % accuracy | %omissions |
| --- | --- | --- | --- | --- | --- | --- |
| S1 | 20 | 2 | ≥30 | ≥40 | - | - |
| S2 | 8 | 2 | ≥40 | ≥50 | - | - |
| S3 | 8 | 5 | | | ≥80 | ≤50 |
| S4 | 4 | 5 | | | ≥80 | ≤50 |
| S5 | 2 | 5 | | | ≥80 | ≤50 |
| Challenges | | | | | | |
| C1 | 2 | 9 | Impulsivity challenge | | | |
| C2 | 1 | 5 | Attention challenge 1 | | | |
| C3 | 0.8 | 5 | Attention challenge 2 | | | |
| C4 | 2 | 5 | Distraction: 1 s white noise within 0.5–4.5 s of ITI | | | |
| C5 | 2 | 7, 9, 11, 13 | Variable ITI: pseudo-random, equal distribution | | | |

outside of the sound attenuating chamber, and a CCTV camera was mounted on the ceiling to monitor behaviour.

## 5-CSRTT training

The 5-CSRTT training protocol was similar to what we described previously (*Grimm et al., 2018*; *van der Veen et al., 2021*). In brief, after initiation of food restriction, mice were accustomed to the reward (strawberry milk, Müllermilch, G) in their home cage and in the operant box (2–3 exposures each). Then, mice were trained on a simplified operant cycle in which all holes of the 5-poke wall were illuminated for an unlimited time, and the mouse could poke into any one of them to illuminate the reward receptacle on the opposite wall and dispense a 40 µl milk reward. Once mice attained at least 30 rewards each in two consecutive sessions, they were moved to the 5-CSRTT.

During 5-CSRTT training, mice transitioned through five stages of increasing difficulty, based on reaching performance criteria in each stage (*Table 2*). The difficulty of each stage was determined by the length of time the stimulus was presented (SD) and the length of the ITI between the end of the previous trial and the stimulus presentation on the next trial.

The ITI was initiated when the subject exited the reward receptacle after collection of a reward or by the end of a timeout period (see below). The ITI was followed by illumination of one hole on the 5-choice wall for the SD determined by the training stage. A poke in the correct port during the stimulus, or during a subsequent 2 s hold period, was counted as a *correct response,* illuminating the reward receptacle and dispensing 20 µl of milk. If the subject either poked into any hole during the ITI (*premature response*), poked into a non-illuminated hole during the SD or hold period (*incorrect response*), or failed to poke during the trial (*omission*), the trial was not rewarded but instead terminated with a 5 s timeout during which the house light was turned off. The relative numbers of each response type were used as performance indicators measuring premature responding [%*premature* = 100 * (number of premature responses)/(number of trials)], sustained attention [accuracy = 100 * (number of correct responses)/(number of correct and incorrect responses)], and lack of participation [*%omissions* = 100 * (number of omissions)/(number of trials)]. In all stages and tests, sessions lasted 30 min and were performed once daily at the same time of day.

Test days with behavioural challenges were interleaved with at least one training day on the baseline stage (stage 5; see *Table 2* for parameters of all stages). For pharmacological validation, atomoxetine (Tomoxetine hydrochloride, Tocris, UK) diluted in sterile saline (0.2 mg/ml) or saline vehicle were injected i.p. at 10 µl/g mouse injection volume 30 min before testing started. For atomoxetine vs. vehicle within-subject comparison, two tests were conducted separated by 1 week, whereby four animals received atomoxetine on the first day, while the other four received vehicle and vice versa for the second day. Effects of challenges (compared to performance on the prior day with baseline training) and atomoxetine (compared to performance under vehicle) were assessed by paired-samples *t*-tests. Behavioural data gathered in the 5-CSRTT was analysed with Excel and SPSS26.0 (IBM Inc, US).

## Vibrissae-based object localisation task

### Animals

Subjects were three female mice expressing the calcium-sensitive protein GCaMP6s in excitatory neurons, derived by mating the floxed Ai94(TITL-GCaMP6s)-D line (Jackson Laboratories; stock number 024742) with the CamKII-tta (Jackson Laboratories; stock number 003010). Animal husbandry and experimental procedures were approved and conducted in accordance with the United Kingdom Animals (Scientific Procedures) Act 1986 under project licence P8E8BBDAD and personal licences from the Home Office.

## Behavioural hardware

Mice were head-fixed on a treadmill fashioned from a 24 cm diameter Styrofoam cylinder covered with 1.5-mm-thick neoprene. An incremental optical encoder (Broadcom HEDS-5500#A02; RS Components) was used in conjunction with a pyControl rotary encoder adapter to monitor mouse running speed. The pole used for object detection was a blunt 18G needle mounted, via a 3d-printed arm, onto a stepper motor (RS PRO Hybrid 535-0467; RS Components). The stepper motor was mounted onto a motorised linear stage (DDSM100/M; Thorlabs) used to move the pole towards and away from the whisker pad

(controlled by a K-Cube Brushless DC Servo Driver [KBD101; Thorlabs]). The pyControl hardware used and the associated hardware definition are shown in *Figure 7—figure supplement 1*.

## Surgery

6- to 10-week-old mice were anaesthetised with isoflurane (0.8–1.2% in 1 l/min oxygen) and implanted with custom titanium headplates for head fixation and 4 mm diameter cranial windows for imaging as described previously (*Chong et al., 2019*). Peri- and postoperative analgesia was used (meloxicam 5 mg/kg and buprenorphine 0.1 mg/kg), and mice were carefully monitored for 7 days post surgery.

## Behavioural training

Following recovery from surgery, mice were habituated to head fixation (*Chong et al., 2019*) prior to training on the vibrissa-based object localisation task as detailed in the 'Results' section. Data were analysed using MATLAB (MathWorks).

## Social decision-making task

### Animals

12 male C57BL6/J mice (Charles River, France) were used, aged 3 months at the beginning of the experiment. Animals were group-housed (four animals per cage) and maintained with ad libitum access to food and water in a 12–12 hr reversed light cycle (lights off at 8 am) at the Animal Facility of the Instituto de Neurociencias of Alicante. Short food restrictions (2 hr before the behavioural testing) were performed in the early phases of individual training to increase motivation for food-seeking behaviour, otherwise animals were tested with ad libitum chow available in their home cage. All experimental procedures were performed in compliance with institutional Spanish and European regulations, as approved by the Universidad Miguel Hernández Ethics committee.

## Behavioural hardware

The social decision-making task was performed in a double maze, where two animals, the focal and the recipient, would interact and work to obtain food rewards. The outer walls of the double maze were of white laser-cut acrylic. Each double maze was divided by a transparent and perforated wall creating the individual mazes for each mouse. For each individual maze, inner walls separating central choice and side reward areas contained the mechanisms for sliding doors, 3D-printed nose-pokes, and position detectors. These inner walls were made of transparent laser-cut acrylic in order to allow visibility of the animal in the side arms of the maze. Walls of the central choice area were frosted to avoid reflections that could interfere with automated pose estimation of the interacting animals in this area.

 Each double T-maze behavioural setup was positioned inside a custom-made sound isolation box, with an infrared-sensitive camera (PointGrey Flea3-U3-13S2M CS, Canada) positioned above the maze to track the animals' location. The chamber was illuminated with dim white light (4 lux) and infrared illumination located on the ceiling of the sound attenuating chamber. The pyControl hardware configuration and associated hardware definition file are shown in *Figure 8—figure supplement 1*. Food pellet rewards were dispensed using pellet dispensers made of 3D-printed and laser-cut parts actuated by a stepper motor (NEMA 42HB34F08AB, e-ika electrónica y robótica, Spain) controlled by a pyControl stepper driver board, placed outside the sound isolation box and delivering the pellets to the 3D-printed food receptacles through a silicon tube. Design files for the pellet dispenser and receptacles are at https://github.com/MarquezLab/Hardware; *Marquez, 2021*. The sliding doors that control access to the side arms were actuated by pneumatic cylinders (Cilindro ISO 6432, Vestonn Pneumatic, Spain) placed below the base of the maze, providing silent and smooth horizontal movement of the doors. These were in turn controlled via solenoid valves (8112005201, Vestonn Pneumatic) interfaced with pyControl using an optocoupled relay board (Cebek-T1, Fadisel, Spain). The speed of the opening/closing of the doors could be independently regulated by adjusting the pressure of the compressed air to the solenoid valves.

## Behavioural training

Individual training and social decision-making protocols are described in the 'Results' section. All behavioural experiments were performed during the first half of the dark phase of the cycle. Data were

analysed with Python (Python Software Foundation, v3.6.5), and statistical analysis was performed with IBM SPSS Statistics (version 26).

## Acknowledgements

TA thanks current and former members of the Champalimaud hardware and software platforms; Jose Cruz, Ricardo Ribeiro, Carlos Mão de Ferro, and Matthieu Pasquet for discussions and technical assistance, and Filipe Carvalho and Lídia Fortunato of Open Ephys Production Site for hardware assembly and distribution. CM thanks Victor Rodriguez for assistance in developing the social decision-making apparatus. MP and MK thank Dr Ana Carolina Bottura de Barros and Dr Severin Limal for assistance with the Vibrissae-based object localisation task.

## Additional information

### Competing interests

Thomas Akam: Consulting contract with Open Ephys Production Site. The other authors declare that no competing interests exist.

### Funding

| Funder | Grant reference number | Author |
| --- | --- | --- |
| Wellcome Trust | WT096193AIA | Thomas Akam |
| Wellcome Trust | 214314/Z/18/Z | Thomas Akam Mark E Walton |
| Wellcome Trust | 202831/Z/16/Z | Mark E Walton |
| Ministerio de Ciencia e Innovación | RTI2018-097843-B-100 and RYC-2014-16450 | Cristina Márquez |
| Ministerio de Ciencia e Innovación | SEV-2017-0723 | Cristina Márquez |
| Generalitat Valenciana and European Union | ACIF/2019/017 | Joan Esteve-Agraz |
| Else-Kroner-Fresenius-Foundation/German-Scholars-Organization | GSO/EKFS 12 | Dennis Kätzel |
| Deutsche Forschungsgemeinschaft | KA 4594/2-1 | Dennis Kätzel |
| Wellcome Trust | 109908/Z/15/Z | Mariangela Panniello |
| Human Frontiers Science Programme | RGY0073/2015 | Michael M Kohl |
| National Institutes of Health | 5U19NS104649 | Rui M Costa |
| H2020 European Research Council | 617142 | Rui M Costa |

The funders had no role in study design, data collection and interpretation, or the decision to submit the work for publication.

### Author contributions

Thomas Akam, Conceptualization, Formal analysis, Funding acquisition, Investigation, Software, Writing - original draft; Andy Lustig, James M Rowland, Software, Writing – review and editing; Sampath KT Kapanaiah, Joan Esteve-Agraz, Mariangela Panniello, Investigation, Writing – review and editing; Cristina Márquez, Dennis Kätzel, Conceptualization, Resources, Supervision, Writing – review and editing; Michael M Kohl, Mark E Walton, Conceptualization, Funding acquisition, Resources,

Supervision, Writing – review and editing; Rui M Costa, Conceptualization, Funding acquisition, Supervision, Writing – review and editing

### Author ORCIDs
Thomas Akam ![ORCID] http://orcid.org/0000-0002-1810-0494
Andy Lustig ![ORCID] http://orcid.org/0000-0002-1358-8363
James M Rowland ![ORCID] http://orcid.org/0000-0001-9140-8260
Cristina Márquez ![ORCID] http://orcid.org/0000-0003-1948-2727
Michael M Kohl ![ORCID] http://orcid.org/0000-0002-2566-5426
Rui M Costa ![ORCID] http://orcid.org/0000-0003-1707-1051
Mark E Walton ![ORCID] http://orcid.org/0000-0003-0117-2894

### Ethics

The 5-CSRTT experiment was performed in accordance to the German Animal Rights Law (Tierschutzgesetz) 2013 and approved by the Federal Ethical Review Committee (Regierungspräsidium Tübingen) of Baden-Württemberg. The Vibrissae-based object localisation experiment was conducted in accordance with the United Kingdom Animals (Scientific Procedures) Act 1986 under project license P8E8BBDAD and personal licenses from the Home Office. The Social decision making experiment was performed in compliance with institutional Spanish and European regulations, as approved by the Universidad Miguel Hernández Ethics committee.

### Decision letter and Author response
Decision letter https://doi.org/10.7554/eLife.67846.sa1
Author response https://doi.org/10.7554/eLife.67846.sa2

## Additional files

### Supplementary files
• Transparent reporting form

### Data availability
pyControl task files for all experiments, and data and analysis code for the performance validation experiments (Figure 5), are included in the manuscript's data repository (https://github.com/pyControl/manuscript; copy archived at swh:1:rev:6be55c29ec0520a61099d25e944b30c9a3bede9b).

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
