## [Editor Report]

The importance of carefully-considered animal behavior to systems neuroscience cannot be overstated. Despite this, flexible tools for carefully monitoring and controlling behavioral apparatuses have often required significant new development by individual laboratories. The open source pyControl software and hardware toolbox is an excellent exemplar of a robust and reliable platform for experiments, with a simple interface, good performance, excellent documentation, and a growing an engaged user community. This work benchmarks and documents pyControl and hopefully will serve as a useful introduction to an even broader community.

---

## [Decision Letter]

**Decision letter after peer review:**

Thank you for sending your article entitled "pyControl: Open source, Python based, hardware and software for controlling behavioural neuroscience experiments" for peer review at *eLife*. Your article is being evaluated by 3 peer reviewers, one of whom is a member of our Board of Reviewing Editors, and the evaluation is being overseen by Kate Wassum as the Senior Editor.

The reviewers were unanimous in recognizing that the pyControl hardware/software suite is a useful tool. Indeed, the impressive list of current users makes that point. However, we felt that in order for the manuscript to merit publication, it needed to be more clear and transparent in presentation such that readers who are not current users would be convinced (by the material presented) that its usability – when compared with alternatives – is such that they should adopt it. By usability, we mean both (i) how easy it is to use for actual tasks and (ii) whether there are scenarios where task parameters or coding naivety make it fragile.

*Reviewer #1:*

Positives:

Most behavior systems take one of two broad approaches to the system configuration: run the task logic and data I/O through the host computer, which is connected to peripherals; or run the task logic and data I/O through an integrated external breakout board and microcontroller. Here, the authors have opted for the latter approach, arguing that it confers several advantages: 1) inexpensiveness, 2) flexibility, 3) user-friendliness, and 4) parallelism.

Certainly, microcontrollers (and their commercial or custom breakout boards) are a less expensive counterpart to desktop computers, albeit at the expense of lower processing power and memory. For experiments that require more computation, this platform may not be the best fit, which the authors acknowledge in their discussion. However, for many experiments, a microcontroller is sufficient, and thus the approach pyControl takes is reasonable.

The flexibility of the system is apparent in both the hardware and software. The custom breakout board offers a mixture of digital and analog I/O ports, as well as power supply lines for different peripheral devices. On the software side, the choice of state machine logic allows it to be broadly applicable. The ability to save both input/output streams and internal variables with timestamps is important.

Major Concern:

pyControl is a combination of hardware, API code, "task code", and control-computer code. For publication, a paper should go beyond merely showing a few tasks where code was used – it needs to not only demonstrate utility, but also document the system in ways beyond a user guide. While demonstrating the efficacy and reliability of the "task code" is the primary goal of this paper, it fails to do so convincingly.

0) Given the advantage of microcontroller-based behavioral control, the primary question facing this paper is: "Why should a lab deal with pyControl rather than rolling their own solution?" It seems like the easiest way to demonstrate this would be to show, e.g., an Arduino solution and a pyControl solution side by side so that it can be seen how obviously easier it is.

1) Figure 1 is confusing to the reader where it appears:

Figure 1 shows an example of running a simple "blink LED" task, and while the right panel is a nice visual complement to the task code, the actual task definition code on the left is at first unclear. Where is the "button_press" event (or any other event) defined when it becomes input to the task functions? Where do Digital_input() and Digital_output() come from? How are "entry" and "exit" events defined? (As an aside, it is usually best to avoid "import *" statements, as they can at best be confusing and at worst cause namespace conflicts. In this case, it is unclear which of the other module classes/functions came from "pyControl.utility" vs. "devices".) Perhaps it would be best to first introduce the overall organization of the hardware (Figure 2, 3), and then explain the task logic that is running on the microcontroller, which itself requires an understanding of how hardware is defined in the first place.

2) Hardware is given short shrift. The hardware definition examples in the supplementary figures seem superfluous. The configuration diagram are useful visuals for understanding how components are wired together, but the code underneath is mostly repetitive and can be left for the interested reader to find in the online documentation. Critically, what is missing is a broader explanation of how hardware devices are defined – both from an API perspective and in terms of the physical connection – that would allow the user to design their own.

3) Links to documentation do not belong in a scientific paper: Choosing a higher-level language like (micro)python makes the system user-friendly and separate from proprietary software like MATLAB. Additionally, the use of a GUI facilitates understanding of the task code and logic. The authors have clearly spent time building an extensive online documentation that is valuable for reaching a broad user base. However, the assumption of the reviewer is that our journal is more reliably archival than random sites on the internet. So anything that is relevant to understanding the paper should be included. (It may be possible to archive the code base along with the paper, which would solve this issue.)

4) There is significant concern about performance and reliability. What happens if two state transitions are triggered simultaneously? There is a description of an event queue: does the queue generation code automatically infer whether queued events are still valid – are race conditions possible? (Simple example – a triggered state transition that should occur if a timer has not elapsed, but a different set of states occurs if the timer has elapsed. What happens if the timer elapses and the trigger occur nearly simultaneously?)

5) There is concern about the choice of Micropython rather than a compiled language. In particular, it is unclear whether the performance testing adequately explored garbage collection. For example, if a naive user instantiates a timer in a state that is executed repeatedly, garbage collection will be regularly required to delete the relevant heap memory. If garbage collection is slow, this could result in unexpected loss of responsiveness or missed logging of data.

6) From the text, it appears that data are timestamped at 1 kHz, but the figures depict microsecond resolution. How fast can events occur? Most experiments will have events on the scale of second or 100s of milliseconds, but a naive user might instantiate a state tracking licks, which can occur much more rapidly when a mouse licks in a bout.

*Reviewer #2:*

The authors have presented a novel hardware/software solution called pyControl for coordinating common types of inputs and outputs in rodent behavioral experiments. It can be also be used to program tasks with other model organisms, including humans, but many of the peripheral devices (such as the nose-poke and lickometer) are specifically designed for use with rodents.

A key advantage of pyControl over other solutions in this space is the ability to program both the software and the embedded firmware in the same high-level language, Python. This allows users to have precise control of the system behavior without the need to learn a lower-level programming language, such as C++. It is also a highly cost effective solution, especially in comparison to closed-source platforms with similar functionality.

A major selling point of the pyControl ecosystem is the wide range of commercially available peripheral devices. For those considering adopting pyControl for their own work, it would be helpful if this manuscript included a more thorough characterization of the existing peripherals. Some of these devices are described in the "application examples" section, but there is almost no information about their specifications. It would also be useful to see an outline of the steps required to build a custom peripheral device.

To characterize the performance of pyControl, the authors have measured its response latency and timing accuracy under both "low load" and "high load" conditions. Even under conditions of high load (monitoring two highly active digital inputs and sampling two analog inputs at 1 kHz), the overwhelming majority of latencies were <1 ms, which is sufficient for most applications. In cases where very precisely timed stimuli are required, pyControl can be easily configured to send triggers to external devices.

Overall, pyControl is a welcome addition to the ever-growing list of tools for coordinating behavioral tasks. Labs that currently rely on undocumented, DIY solutions may find that pyControl provides the impetus to switch to a standardized, open-source platform. And researchers that depend on closed-source tools have the opportunity to save many thousands of dollars per year by using pyControl for their future rigs.

This manuscript will be a great resource for researchers that are considering adopting pyControl. I have a few suggestions for making it even better:

I'd like to see a more candid discussion of the limitations of pyControl for various types of experiments. For example, running everything on a microcontroller precludes the display of complex visual stimuli. Therefore, pyControl would not be suitable for most visual physiology experiments. Similarly, the pyControl Audio Player peripheral can only deliver stimuli up to 48 kHz, which limits its utility for rodent auditory physiology. pyControl of course includes general-purpose output ports that can be used to trigger any auxiliary devices, so these limitations could be overcome with some customization.

The authors should include more details about the pyboard microcontroller, since this is an essential component of the platform. What are the specifications, approximate cost, etc.? Is there a plan to take advantage of the pyBoard's wireless capabilities? This could be helpful for coordinating massively parallel experiments.

The "Karpova Lab" is mentioned out of context. Readers will likely want more information about what this means.

Since Bonsai does not itself encompass any hardware interfaces, a more accurate comparison would be with the combination of Bonsai and Harp (https://www.cf-hw.org/harp). Unfortunately, Harp doesn't have a publication associated with it, nor much online documentation. But the main advantages are shared timestamps (within 44 µs) across all devices, and tight integration with Bonsai software. The authors should mention the availability of Harp in their discussion of Bonsai.

Another behavior control platform that should be cited is Autopilot (https://www.biorxiv.org/content/10.1101/807693v1), which looks promising for certain applications, despite its relative immaturity.

A detailed characterization of closed-source commercial hardware is beyond the scope of this manuscript, but the authors should consider presenting at least a high-level comparison of these tools (in terms of order-of-magnitude cost, available features, and customizability). The discussion seems to be targeted at users choosing between various open-source solutions, but convincing researchers to adopt pyControl in favor of closed-source hardware is a much more important goal.

*Reviewer #3:*

Akam et al. detail an open-source system called pyControl, that is intended to be used to control behavioral neuroscience experiments. The authors nicely highlight the current problem of reproducibility in behavioral neuroscience, describing the difficulty of standardizing or understanding code and behavioral tasks across labs. It is quite clear the project is setting new standards, especially in the open-source neuroscience field, by addressing many pitfalls of previous behavioral systems and platforms. Thus, the major strength of the paper is the open-source manner with which the authors detail their system. The hardware and software provided in an open manner allows for researchers to streamline their behavioral tasks in a cost-effective manner while still maintaining the flexibility needed to perform specific experiments. The authors provide rather straightforward documentation of how to program and run behavioral experiments using pyControl software and the related GUI, as well as how the hardware is generally set up. The authors then validate the use of pyControl in three different behavioral tasks that are widely used in neuroscience research.

Strengths: The pyControl system itself serves the purpose of having one streamlined behavioral setup that can provide reproducibility across research labs, the system can seemingly be used by "novice" researchers with a beginner level skill set in both hardware and computer programming, and comes at a cheaper price than commercial (or other open source) behavior platforms. The project's website, user forum, and github are evidence of a well-supported project that is already being used across many research labs. The documentation is written in a tutorial-style manner, providing many examples and references between hardware and software, and making it easy to follow.

The authors additionally show convincing evidence of the robustness of the pyControl system with respect to the different behavioral paradigms it is capable of supporting. Many other behavior systems are limited in what tasks they can be used with, and the evidence of a flexible steup across different experiments in the manuscript is quite convincing. An added benefit of the system is its ability to time synchronize to electrophysiological or imaging platforms for optimal data analysis. The system, as a whole, is a "one-stop shop" for programming, running, and visualizing experiments in real time, and then open-source nature of the system allows researchers to (mostly) pick and choose how they'd like to utilize the system. The project has already gained steam with many different labs using it currently, and has the potential for improving rigor and robustness of understanding behavioral data across research labs.

Weaknesses: While the manuscript claims that pyControl is a flexible device, it is not clear how other devices could interact with the pyControl hardware. Many labs already have behavior devices (such as lickometers or solenoids or odor delivery ports, for example), either custom-built or from commercial vendors that already work well, but the researchers may instead be looking for a way to run their tasks in a simpler or well-documented manner. It is stated that it is 'straightforward' to make a pyControl compatible device but this is unclear within the manuscript how much work this would take, or the ease of using one's own devices. While the system is intended to have everything that anyone needs, the benefit of an open-source framework is that people are allowed to flexibly choose which aspects might work optimally for their research question. It would be best to at least detail how people could adapt their own devices. In the same realm, it is a bit unclear of the extent of capabilities of the system, with respect to data processing or memory storage, running multiple animals at once, or platform dependency, for example.

[Editors' note: further revisions were suggested prior to acceptance, as described below.]

Thank you for resubmitting your work entitled "Open source, Python based, hardware and software for controlling behavioural neuroscience experiments." for further consideration by *eLife*. Your revised article has been evaluated by Kate Wassum (Senior Editor) and a Reviewing Editor.

The manuscript has been improved but there are some remaining issues that need to be addressed, as outlined below from Reviewer 1. The reviewers agreed that it is important that the manuscript be largely self-contained.*Reviewer #1:*

A journal paper is meant to document the past for reproduction, not advertise some potential future. The ephemeral nature of online documentation is not conducive to this. However, assuming the documentation is archived with the paper, I am satisfied that this can be addressed.

There is no reason not to adequately document the RJ-45 connector interface within the paper. It represents the current state of the hardware as opposed to some hypothetical future flexibility, requires minimal text, and represents a clear critical specification for users who wish to design their own modules. I see no reason for the authors not to do this.

The authors link to their help documentation to reply to our concern about "from pyControl.utility import *" and "from devices import *". This documentation essentially admits that unless one has completely understood the pyControl codebase (e.g., one of the developers) to know all of the functions and objects in the namespace, one is hopeless to avoid having collisions. This seems to directly contradict the claim that they have built a broadly useful tool.

*Reviewer #2:*

The revised manuscript is much improved, and the authors have addressed all of my concerns. This manuscript, in combination with the online documentation, will be an excellent resource for learning about pyControl.

*Reviewer #3:*

The authors have adequately responded to the reviews and provided a thorough and improved manuscript. I do not believe any additional work is required to the updated manuscript. I also agree with the authors that, as discussed around lines 673 in the author comments document, it is not necessary to include the additional panels for figure 7 as it would be repetitive to what is already in the figure.

---

## [Author Response]

Reviewer #1:Positives:Most behavior systems take one of two broad approaches to the system configuration: run the task logic and data I/O through the host computer, which is connected to peripherals; or run the task logic and data I/O through an integrated external breakout board and microcontroller. Here, the authors have opted for the latter approach, arguing that it confers several advantages: 1) inexpensiveness, 2) flexibility, 3) user-friendliness, and 4) parallelism.Certainly, microcontrollers (and their commercial or custom breakout boards) are a less expensive counterpart to desktop computers, albeit at the expense of lower processing power and memory. For experiments that require more computation, this platform may not be the best fit, which the authors acknowledge in their discussion. However, for many experiments, a microcontroller is sufficient, and thus the approach pyControl takes is reasonable.The flexibility of the system is apparent in both the hardware and software. The custom breakout board offers a mixture of digital and analog I/O ports, as well as power supply lines for different peripheral devices. On the software side, the choice of state machine logic allows it to be broadly applicable. The ability to save both input/output streams and internal variables with timestamps is important.

We thank the reviewer for their close reading of the manuscript and support for the high level design choices made in pyControl.

Major Concern:pyControl is a combination of hardware, API code, "task code", and control-computer code. For publication, a paper should go beyond merely showing a few tasks where code was used – it needs to not only demonstrate utility, but also document the system in ways beyond a user guide. While demonstrating the efficacy and reliability of the "task code" is the primary goal of this paper, it fails to do so convincingly.

We agree with the reviewer that the robustness and reliability of the system are paramount, and hope we have addressed their concerns through the additional information and validation experiments discussed below.

0) Given the advantage of microcontroller-based behavioral control, the primary question facing this paper is: "Why should a lab deal with pyControl rather than rolling their own solution?" It seems like the easiest way to demonstrate this would be to show, e.g., an Arduino solution and a pyControl solution side by side so that it can be seen how obviously easier it is.

While we understand where the reviewer is coming from, we think that this suggestion to compare an Arduino based solution and pyControl is less straightforward than it sounds. pyControl is a complete system for controlling behavioural experiments comprising a task definition syntax and associated framework code, a graphical interface for running experiments on many setups in parallel and visualising what is happening in real-time, hardware modules for building setups, plus the extensive user documentation needed to make it accessible to the broader community. It is the *combination* of these features which makes pyControl valuable.

It is clearly not feasible to develop an Arduino based system with equivalent functionality to address this comment. It may be that the reviewer is imagining a narrower comparison of code to implement a single basic task on an Arduino with code to implement an equivalent task in pyControl. If the editor and reviewers believe this would be valuable we are willing to do it, but we are not sure it would be very informative:

First, because getting a minimal task implementation working is really not the same thing as having a practical experiment control system. At a bare minimum, the system requires additional code on the computer to acquire the data coming of the board and save it to disk. To be practical, a system needs substantially more functionality than this, e.g. the ability to select which task to run on which setup, load the appropriate task code, configure task variables, start and stop the task, visualise what is going on during the experiment, etc. The task implementation is not independent of this functionality as it needs to interact with it. A narrow comparison of a minimal task implementation therefore gives a potentially very misleading picture of the amount of work needed to get a viable system up and running from scratch.

Second, the pyControl framework implements functionality that we are not sure would be possible to implement on an Arduino, and we certainly would not know how to, e.g. streaming data from analog inputs and rotary encoders, and triggering task events when these cross thresholds, in parallel to processing digital inputs and state machine code. Again, showing a minimal Arduino example task that does not implement this functionality could be misleading

1) Figure 1 is confusing to the reader where it appears:Figure 1 shows an example of running a simple "blink LED" task, and while the right panel is a nice visual complement to the task code, the actual task definition code on the left is at first unclear. Where is the "button_press" event (or any other event) defined when it becomes input to the task functions? Where do Digital_input() and Digital_output() come from? How are "entry" and "exit" events defined? (As an aside, it is usually best to avoid "import *" statements, as they can at best be confusing and at worst cause namespace conflicts. In this case, it is unclear which of the other module classes/functions came from "pyControl.utility" vs. "devices".) Perhaps it would be best to first introduce the overall organization of the hardware (Figure 2, 3), and then explain the task logic that is running on the microcontroller, which itself requires an understanding of how hardware is defined in the first place.

We think there is a tension here and elsewhere between providing detailed technical information – e.g. about every element of the example task code shown in figure 1, and maintaining the readability and length of the manuscript. Our approach to this has been a division of labour between the manuscript and online documentation, whereby the manuscript gives a high level overview of the design principles and their rationale, while detailed technical information primarily relevant to users or developers resides in the docs. The documentation currently runs to ~17000 words, and while we think this level of detail is valuable, it is clearly not possible in a manuscript. As the reviewer notes below, this necessitates that the documentation is robustly archived, as it is a key component of the overall project, and this is already implemented (see below).

We therefore think that while it is useful to clarify these points, it is probably best to do this in the documentation. We have added a link to the relevant documentation from the figure legend and have added the section "Where do phControl functions come from" to the programming tasks docs addressing points raised by the reviewer that were not already covered in the docs:

2) Hardware is given short shrift. The hardware definition examples in the supplementary figures seem superfluous. The configuration diagram are useful visuals for understanding how components are wired together, but the code underneath is mostly repetitive and can be left for the interested reader to find in the online documentation. Critically, what is missing is a broader explanation of how hardware devices are defined – both from an API perspective and in terms of the physical connection – that would allow the user to design their own.

Thanks for this suggestion – a similar point was also raised by reviewer 3. We agree that it would be useful to provide more information about how users can integrate custom external hardware with pyControl and define new devices, and think that the best way to do this is to add in a high level overview in the manuscript as well as detailed information in the documentation. We have added new material to both.

As requested we have removed the hardware definition code from the supplementary figure legends and replaced it with links to the files in the manuscript’s data repository.

New material in manuscript:

“To extend the functionality of pyControl to application not supported by the existing hardware, it is straightforward to interface setups with user created or commercial devices. This requires creating an electrical connection between the devices and defining the inputs and outputs in the hardware definition. Triggering external hardware from pyControl, or task events from external devices, is usually achieved by connecting the device to a BNC connector on the breakout board, and using the standard pyControl digital input or output classes. More complex interactions with external devices may involve multiple inputs and outputs and/or serial communication. In this case the electrical connection is typically made to a behaviour port, as these carry multiple signal lines. A port adapter board, which breaks out an RJ45 connector to a screw terminal, simplifies connecting wires. Alternatively, if more complex custom circuitry is required, e.g. to interface with a sensor, it may make sense to design a custom printed circuit board with an RJ45 connector, similar to existing pyControl devices, as this is more scalable and robust than implementing the circuit on a breadboard. To simplify instantiating devices comprising multiple inputs and outputs, or controlling devices which require dedicated code, users can define a Python class representing the device. These are typically simple classes which instantiate the relevant pyControl input and output objects as attributes, and may have methods containing code for controlling the device, e.g. to generate serial commands. More information is provided in the section “Interfacing with external hardware” in the hardware docs, and the design files and associated code for existing pyControl devices provide a useful starting point for new designs.”

New material in documentation in the section “Interfacing with external hardware”

3) Links to documentation do not belong in a scientific paper: Choosing a higher-level language like (micro)python makes the system user-friendly and separate from proprietary software like MATLAB. Additionally, the use of a GUI facilitates understanding of the task code and logic. The authors have clearly spent time building an extensive online documentation that is valuable for reaching a broad user base. However, the assumption of the reviewer is that our journal is more reliably archival than random sites on the internet. So anything that is relevant to understanding the paper should be included. (It may be possible to archive the code base along with the paper, which would solve this issue.)

We completely agree with the reviewer that as the docs are an important component of the overall system, it is essential that they are robustly archived. This is already implemented; the source files for the documentation are hosted in the pyControl docs repository on Github. The website where the docs are displayed (ReadTheDocs) automatically pulls the latest version from Github each time the repository is updated, ensuring that the archived and served docs are always in sync and fully version controlled. We also note that ReadTheDocs is a very well established service in its own right, which currently hosts documentation for over 240,000 open source projects, see https://readthedocs.org/sustainability/.

We respectfully disagree with the reviewer that links to the documentation should not be in the manuscript. We see the manuscript and documentation as providing complementary information about the system, as discussed above, and therefore think that providing tight integration between them is desirable. However, we will abide by the policy of the journal on this issue.

4) There is significant concern about performance and reliability. What happens if two state transitions are triggered simultaneously? There is a description of an event queue: does the queue generation code automatically infer whether queued events are still valid – are race conditions possible? (Simple example – a triggered state transition that should occur if a timer has not elapsed, but a different set of states occurs if the timer has elapsed. What happens if the timer elapses and the trigger occur nearly simultaneously?)

Performance and reliability are clearly critical. Regarding reliability, we note that pyControl has at this point been used to run many thousands of hours of behaviour across tens of labs and at least 10 different publications and preprints (referenced at line 349 in revised manuscript). The core framework code which handles functionality like processing events and elapsed timers has been largely unchanged for several years now, with recent development work largely focussed on developing and extending GUI functionality. Regarding the specific scenarios outlined above:

What happens if two state transitions are triggered simultaneously?

It is not possible for two state transitions to be triggered simultaneously because events are processed sequentially from the event queue, and during event processing individual lines of code in the state behaviour function are themselves processed sequentially. To further enforce predictable behaviour around state transitions, users are prevented from calling the goto_state function during processing of the entry and exit events that occur during state transitions (attempting to do so gives an informative error message), ensuring that any state entry and/or exit behaviour associated with a state transition has run to completion before another state transition can be triggered.

There is a description of an event queue: does the queue generation code automatically infer whether queued events are still valid?

pyControl has no concept of events being valid or invalid. If an event is in the queue it will be processed, and the consequences of this will depend on the task’s state when the event is processed.

- are race conditions possible? (Simple example – a triggered state transition that should occur if a timer has not elapsed, but a different set of states occurs if the timer has elapsed. What happens if the timer elapses and the trigger occur nearly simultaneously?)

If we understand correctly, the reviewer is asking whether there is any indeterminacy in the order in which an elapsing timer and a digital input will be processed if they occur near simultaneously. As already quantified in the manuscript (Figure 5C,D), there is variation in the duration of timed intervals due to the 1 ms resolution of the framework clock ticks, with standard deviation of 282us. We think in most scenarios this is likely to dominate processing order indeterminacy in this situation. Some additional variability will be introduced by the time taken to process individual events and other unitary framework operations (such as streaming a chunk of data to the computer), as if both a clock tick and external input occur during a single operation, the external input will be processed before any timers elapsing on that tick, due to the priority of framework operations shown in figure 2. Timing accuracy in the ‘high load’ condition, quantified in figure 5D, gives a reasonable worst case scenario for the combined influence of these two factors, resulting in a variation in timed intervals with standard deviation 353us. Therefore while there is some indeterminacy in execution order in this scenario it is on a timescale that is not relevant for behaviour.

5) There is concern about the choice of Micropython rather than a compiled language. In particular, it is unclear whether the performance testing adequately explored garbage collection. For example, if a naive user instantiates a timer in a state that is executed repeatedly, garbage collection will be regularly required to delete the relevant heap memory. If garbage collection is slow, this could result in unexpected loss of responsiveness or missed logging of data.

The use of a high level language on the microcontroller is responsible for many of the strengths of the system in terms of ease of use and flexibility. Indeed reviewer 2 stated that ‘A key advantage of pyControl over other solutions in this space is the ability to program both the software and the embedded firmware in the same high-level language, Python’*.*

However, we agree it is important to understand issues that could be caused by garbage collection. We have performed new validation experiments and added two new panels to figure 5, and associated results text, testing the effect of garbage collection on timers and external inputs.

To address the specific concern that a user might load the system by setting timers in quick succession, causing unresponsiveness due to repeated garbage collection: We tested an extreme example of this where we used timers with 1ms duration to transition continuously between two states at a frequency of 1KHz. In one state a digital output was on and in another off, such that by monitoring the output we could observe when garbage collection occurred due to the short delay it caused in the alternation between states. Garbage collection occurred following 0.6% of state transitions, taking about 3ms each time, such that overall about 2% of the systems time was spent on garbage collection. As events that occur during garbage collection are still processed once it has completed, even this deliberately extreme scenario would have minimal effects on the systems responsiveness. New figure panels 5E,F.

New results text:

“Users who require very tight timing/latency performance should be aware of Micropython’s automatic garbage collection. Garbage collection is triggered when needed to free up memory and takes a couple of milliseconds. Normal code execution is paused during garbage collection, though interrupts (used to register external inputs and update the framework clock) run as normal. pyControl timers that elapse during garbage collection will be processed once it has completed (Figure 5E). Timers that are running but do not elapse during garbage collection are unaffected. Digital inputs that occur during garbage collection are registered with the correct timestamp (Figure 5F), but will only be processed once garbage collection has completed. The only situation where events may be missed due to garbage collection is if a single digital input receives multiple event-triggering edges during a single garbage collection, in which case only the last event is processed correctly (Figure 5F). To avoid garbage collection affecting critical event processing, the user can manually trigger garbage collection at a time when it will not cause problems (see Micropython docs), for example during the inter-trial interval.”

6) From the text, it appears that data are timestamped at 1 kHz, but the figures depict microsecond resolution. How fast can events occur? Most experiments will have events on the scale of second or 100s of milliseconds, but a naive user might instantiate a state tracking licks, which can occur much more rapidly when a mouse licks in a bout.

The system clock tick that updates the current time (used for event timestamps) and determines when timers elapse, runs at 1Khz. However external events are detected by hardware interrupts rather than polling at the clock frequency, hence the 556 ± 17 μs latency under low load conditions. Regarding how fast events can occur; simultaneous events on different digital inputs are fine, as shown in figure 5F (reproduced above) where an input signal was connected to multiple pyControl digital inputs such that events were generated on multiple inputs by each edge of the signal. For events generated by a single digital input, pairs of edges separated by 1ms are processed correctly, unless both edges occur during a single garbage collection (see figure 5F and associated text above). Regarding the maximum continuous event rate that the system can handle, we did some additional testing of this which is reported in the revised manuscript as:

"A final constraint is that as each event takes time to process, there is a maximum continuous event rate above which the framework cannot process events as fast as they occur, causing the event queue to grow until available memory is exhausted. This rate will depend on the processing triggered by each event, but is approximately 960Hz for digital inputs triggering state transitions but no additional processing. In practice we have never encountered this when running behavioural tasks as average event rates are typically orders of magnitude lower and transiently higher rates are buffered by the queue."

Reviewer #2:[…]This manuscript will be a great resource for researchers that are considering adopting pyControl. I have a few suggestions for making it even better:

We thank the reviewer for their positive assessment of the manuscript and system.

I'd like to see a more candid discussion of the limitations of pyControl for various types of experiments. For example, running everything on a microcontroller precludes the display of complex visual stimuli. Therefore, pyControl would not be suitable for most visual physiology experiments. Similarly, the pyControl Audio Player peripheral can only deliver stimuli up to 48 kHz, which limits its utility for rodent auditory physiology. pyControl of course includes general-purpose output ports that can be used to trigger any auxiliary devices, so these limitations could be overcome with some customization.

We had tried to be as straightforward as possible in the discussion about the limitations of the system as well as advantages, but we agree there are some additional points that should be raised about visual and auditory stimuli. We felt this material fitted best in the section detailing hardware, and have added a paragraph that reads (lines 195-203):

“The design choice of running tasks on a microcontroller, and the specific set of devices developed to date, impose some constraints on experiments supported by the hardware. The limited computational resources preclude generating complex visual stimuli, making pyControl unsuitable for most visual physiology in its current form. The devices for playing audio are aimed at general behavioural neuroscience applications, and may not be suitable for some auditory neuroscience applications. One uses the pyboard’s internal DAC for stimulus generation, and hence is limited to simple sounds such as sine waves or noise. Another plays WAV files from an SD card, allowing for diverse stimuli but limited to 44KHz sample rate.”

The authors should include more details about the pyboard microcontroller, since this is an essential component of the platform. What are the specifications, approximate cost, etc.?

We now provide the key specifications (Arm Cortex M4 running at 168MHz with 192KB RAM) at line 168. The pyboards are cheap (£28), but as the cost of the microcontroller is generally a small fraction of the overall system cost, we do not specify it to avoid giving a misleading picture.

Is there a plan to take advantage of the pyBoard's wireless capabilities? This could be helpful for coordinating massively parallel experiments.

The version of the pyboard that we use currently does not have built in wireless capability, though there are other Micropython boards that do. We do not have plans to implement wireless data transmission at this point as our feeling is that implementation and reliability issues are likely to outweigh advantages for most applications, particularly as the system will likely still need to be wired to provide power.

The "Karpova Lab" is mentioned out of context. Readers will likely want more information about what this means.

Fixed in revised manuscript (line 220).

Since Bonsai does not itself encompass any hardware interfaces, a more accurate comparison would be with the combination of Bonsai and Harp (https://www.cf-hw.org/harp). Unfortunately, Harp doesn't have a publication associated with it, nor much online documentation. But the main advantages are shared timestamps (within 44 µs) across all devices, and tight integration with Bonsai software. The authors should mention the availability of Harp in their discussion of Bonsai.

We now mention the HARP hardware in the discussion of Bonsai, saying (lines 575-578):

“Though Bonsai itself is software, some compatible behavioural hardware has been developed by the Champalimaud Foundation Hardware Platform (https://www.cf-hw.org/harp), which offers tight timing synchronisation and close integration with Bonsai, but documentation is currently limited.”

Another behavior control platform that should be cited is Autopilot (https://www.biorxiv.org/content/10.1101/807693v1), which looks promising for certain applications, despite its relative immaturity.

Now cited at line 478.

A detailed characterization of closed-source commercial hardware is beyond the scope of this manuscript, but the authors should consider presenting at least a high-level comparison of these tools (in terms of order-of-magnitude cost, available features, and customizability). The discussion seems to be targeted at users choosing between various open-source solutions, but convincing researchers to adopt pyControl in favor of closed-source hardware is a much more important goal.

We do discuss the issues with commercial behavioural hardware in broad terms in the introduction (lines 44-51), and think that most of the target audience for this system, i.e. behavioural neuroscientists, are acutely aware of them. We would rather not overemphasise pyControl’s low cost relative to commercial hardware, as we think that this speaks for itself, and the flexibility, user friendliness and open nature of the system are equally important.

Reviewer #3:Akam et al. detail an open-source system called pyControl, that is intended to be used to control behavioral neuroscience experiments. The authors nicely highlight the current problem of reproducibility in behavioral neuroscience, describing the difficulty of standardizing or understanding code and behavioral tasks across labs. It is quite clear the project is setting new standards, especially in the open-source neuroscience field, by addressing many pitfalls of previous behavioral systems and platforms. Thus, the major strength of the paper is the open-source manner with which the authors detail their system. The hardware and software provided in an open manner allows for researchers to streamline their behavioral tasks in a cost-effective manner while still maintaining the flexibility needed to perform specific experiments. The authors provide rather straightforward documentation of how to program and run behavioral experiments using pyControl software and the related GUI, as well as how the hardware is generally set up. The authors then validate the use of pyControl in three different behavioral tasks that are widely used in neuroscience research.Strengths: The pyControl system itself serves the purpose of having one streamlined behavioral setup that can provide reproducibility across research labs, the system can seemingly be used by "novice" researchers with a beginner level skill set in both hardware and computer programming, and comes at a cheaper price than commercial (or other open source) behavior platforms. The project's website, user forum, and github are evidence of a well-supported project that is already being used across many research labs. The documentation is written in a tutorial-style manner, providing many examples and references between hardware and software, and making it easy to follow.The authors additionally show convincing evidence of the robustness of the pyControl system with respect to the different behavioral paradigms it is capable of supporting. Many other behavior systems are limited in what tasks they can be used with, and the evidence of a flexible steup across different experiments in the manuscript is quite convincing. An added benefit of the system is its ability to time synchronize to electrophysiological or imaging platforms for optimal data analysis. The system, as a whole, is a "one-stop shop" for programming, running, and visualizing experiments in real time, and then open-source nature of the system allows researchers to (mostly) pick and choose how they'd like to utilize the system. The project has already gained steam with many different labs using it currently, and has the potential for improving rigor and robustness of understanding behavioral data across research labs.

We thank the reviewer for their positive comments about the manuscript and system.

Weaknesses: While the manuscript claims that pyControl is a flexible device, it is not clear how other devices could interact with the pyControl hardware. Many labs already have behavior devices (such as lickometers or solenoids or odor delivery ports, for example), either custom-built or from commercial vendors that already work well, but the researchers may instead be looking for a way to run their tasks in a simpler or well-documented manner. It is stated that it is 'straightforward' to make a pyControl compatible device but this is unclear within the manuscript how much work this would take, or the ease of using one's own devices. While the system is intended to have everything that anyone needs, the benefit of an open-source framework is that people are allowed to flexibly choose which aspects might work optimally for their research question. It would be best to at least detail how people could adapt their own devices. In the same realm, it is a bit unclear of the extent of capabilities of the system, with respect to data processing or memory storage, running multiple animals at once, or platform dependency, for example.

We agree that it is important to make clear how users can interface pyControl with external hardware and have added new material to both the manuscript (lines 204-224) and documentation explaining this. As this issue was also raised by reviewer 1, this material is reproduced above (lines 137-218 in this document) in response to their comment.

[Editors' note: further revisions were suggested prior to acceptance, as described below.]

Reviewer #1:A journal paper is meant to document the past for reproduction, not advertise some potential future. The ephemeral nature of online documentation is not conducive to this. However, assuming the documentation is archived with the paper, I am satisfied that this can be addressed.

We now include a PDF version of the documentation in supplementary material to ensure that it is archived with the paper.

There is no reason not to adequately document the RJ-45 connector interface within the paper. It represents the current state of the hardware as opposed to some hypothetical future flexibility, requires minimal text, and represents a clear critical specification for users who wish to design their own modules. I see no reason for the authors not to do this.

We now include a new table detailing the RJ-45 connector behaviour port interface.

The authors link to their help documentation to reply to our concern about "from pyControl.utility import *" and "from devices import *". This documentation essentially admits that unless one has completely understood the pyControl codebase (e.g., one of the developers) to know all of the functions and objects in the namespace, one is hopeless to avoid having collisions. This seems to directly contradict the claim that they have built a broadly useful tool.

We thank the reviewer for their persistence on this issue, as it has led us to improve the framework code for this revision. This said, we think their above comment somewhat overstates the severity of the issue, as in the six years we have been providing technical support to pyControl users we have yet to encounter a single case in which a name collision has caused a problem. Nonetheless, we agree it would be useful to allow users to use only named imports if they prefer. This was not possible with the previous version of the codebase as some functions were patched into the task definition file after it was imported by the framework, rather than being imported into it by the user. We have now refactored the framework code such that all pyControl-specific functions, variables and classes are imported by the user into the task file. The docs now describe both the use of * imports and named imports, and we provide example task files using both approaches so users can compare the resulting code and decide for themselves which they prefer.

The docs still treat using * import for the *pyControl.utility* module as the default, as this results in a less verbose, and we think clearer, task definition syntax. The risk of an issue arising due to a name collision is very low: The imported functions will not overwrite user defined functions as the import is at the top of the file. If users overwrite a function in the imported module that they were not aware of, this will not cause a problem because these functions are only ever called by the user.

The modified section is in the “Imports” of the docs.